# Hallucination Detection on a Budget: Efficient Bayesian Estimation of Semantic Entropy

**Kamil Ciosek** *kamilc@spotify.com*
*Spotify*

**Nicolò Felicioni** *nicolof@spotify.com*
*Spotify*

**Sina Ghiassian** *sinag@spotify.com*
*Spotify*

**Reviewed on OpenReview:** *https://openreview.net/forum?id=j2N2RuNdbC*

## Abstract

Detecting whether an LLM hallucinates is an important research challenge. One promising way of doing so is to estimate the semantic entropy (Farquhar et al., 2024) of the distribution of generated sequences. We propose a new algorithm for doing that, with two main advantages. First, due to us taking the Bayesian approach, we achieve a much better quality of semantic entropy estimates for a given budget of samples from the LLM. Second, we are able to tune the number of samples adaptively so that 'harder' contexts receive more samples. We demonstrate empirically that our approach systematically beats the baselines, requiring only 53% of samples used by Farquhar et al. (2024) to achieve the same quality of hallucination detection as measured by AUROC. Moreover, quite counterintuitively, our estimator is useful even with just one sample from the LLM.

## 1 Introduction

Detecting hallucinations in LLMs is a task of huge practical significance (Ji et al., 2023). An important subset of hallucinations, called 'confabulatory', amounts to the model making up confabulations or statements with made-up meanings (Filippova, 2020; Maynez et al., 2020). Recently, semantic entropy (Farquhar et al., 2024) has been introduced as an important indicator for detecting if a model exhibits this type of hallucination. Semantic Entropy is based on two principles. The first one is to measure a type of *Shannon entropy* of the sequences generated by a model, reflecting the idea that large entropy indicates confounding or a lack of knowledge. The second principle is to do the measurement in the space of *meanings* rather than directly operating on raw token sequences. By doing so, one can leverage the insight that in many cases, distinct token sequences can have the same meaning. It turns out that combining these insights to estimate semantic entropy and then thresholding on it value amounts to a highly competitive hallucination detection method (Farquhar et al., 2024).

While semantic entropy is a state-of-the art method of hallucination detection, computing it has a high cost. It first requires the generation of several independent answers to the same question and then a quadratic number of calls to the function determining if two meanings are the same. In fact, the work of Farquhar et al. (2024) used ten generations per prompt, which is prohibitively expensive in many practical cases. We address this bottleneck by making semantic entropy estimation much cheaper. We achieve this by leveraging insights from Bayesian literature about entropy estimation (Wolpert & Wolf, 1994; Hausser & Strimmer, 2009; Archer et al., 2014), building up a probabilistic belief about the underlying distribution over meanings and reasoning about how the belief in the space of meaning distributions affects the belief in the space of possible values of the entropy. We further study a novel, adaptive, setting, where 'harder' prompts are

afforded a larger budget of samples. In this setting, the efficiency of our estimator can be increased even further.

**Contributions**   We develop a new system for measuring semantic entropy, based on Bayesian principles. Compared to the work of Farquhar et al. (2024), we outperform all other ways of measuring semantic entropy, reducing the sample complexity measured as the number of LLM generations needed to achieve the same performance by 47% on average across datasets. We also release several datasets for semantic entropy estimation, enabling researchers without access to GPU resources to work on even better estimators.

## 2   Preliminaries

**Language Generation**   Denote with $X$ the set of prompts[1] the LLM can be asked to respond to. One instance[2] of a prompt $\mathbf{x} \in X$ would be

$$\mathbf{x} = \text{`Where is the Eiffel Tower?'}$$

Denote with $S_{\mathbf{x}}$ the set of all reply sequences given a prompt $\mathbf{x}$.[3]   Denote the probability of the LLM generating a sequence $s \in S_{\mathbf{x}}$ in response to the prompt $\mathbf{x} \in X$ with $p(\mathbf{s}|\mathbf{x})$. One possible continuation is

$$\mathbf{s} = \text{`It's Paris.'}$$

We model both $\mathbf{s}$ and $\mathbf{x}$ as random variables, denoting them in bold.

**Meanings**   Semantic entropy is always conditioned on a prompt. We are going to consider a given generic context $\mathbf{x}$. Denote the set of meanings (also known as meaning classes or semantic classes) with $M_{\mathbf{x}}$. The set $M_{\mathbf{x}}$ is a partition of the set of $S_{\mathbf{x}}$.[4] We assume that the set of meanings is finite (although we do not necessarily know its cardinality). Denote with

$$f^{\mathbf{x}}(\mathbf{s}) : S_{\mathbf{x}} \to M_{\mathbf{x}}$$

the function that determines the meaning of the sequence $\mathbf{s}$ in context $\mathbf{x}$. While $f^{\mathbf{x}}$ is typically implemented using calls to an entailment oracle,[5] we abstract away this implementation detail in this paper. For a random sequence $\mathbf{s} \sim p(\cdot|\mathbf{x})$, we consider the random variable

$$\mathbf{m} = f^{\mathbf{x}}(\mathbf{s}).$$

**Semantic Entropy**   The semantic entropy corresponding to the context $\mathbf{x}$ is defined as the Shannon entropy of the random variable $\mathbf{m}$:

$$\mathrm{SE}_{\mathbf{x}} = \mathbb{H}[\mathbf{m}].$$

In the remainder of the paper, we will occasionally drop the subscript $\mathbf{x}$ where the dependence on the context $\mathbf{x}$ is obvious. Semantic entropy is useful for detecting hallucinations because of the following crucial observation (Farquhar et al., 2024).

> Higher values of $\mathrm{SE}_{\mathbf{x}}$ imply the model's response to prompt $\mathbf{x}$ is more likely to be a hallucination.

---

[1] Occasionally, the notion of a 'context' is used in addition to the prompt, to model the phenomenon that the same prompt can have different meanings in different contexts. To keep our notation simple, we don't explicitly model contexts. However, if one wants to generalize our results to contexts, it can be done by considering $\mathbf{x}$ to be a context-prompt tuple.

[2] We use examples borrowed from Farquhar et al. (2024).

[3] Typically, both $X$ and $S_{\mathbf{x}}$ are the set of natural language sequences. However, whether $X = S_{\mathbf{x}}$ is immaterial for this paper.

[4] The term 'partition' is used in the mathematical way so that a sequence $s \in S_x$ always has exactly one meaning.

[5] The entailment oracle is a function that can tell if two meanings are distinct.

**The Estimation Problem**   The aim of this paper is to estimate semantic entropy from a finite dataset, based on $N$ calls to the target LLM. For a given context $\mathbf{x}$, our samples are represented as a list of independently generated sequences

$$\mathbf{s}_1, \ldots, \mathbf{s}_N \sim p(\cdot|\mathbf{x}).$$

For each sequence, we can determine its meaning, obtaining the corresponding list of meanings

$$\mathbf{m}_1, \ldots, \mathbf{m}_N, \text{ where } \mathbf{m}_i = f^{\mathbf{x}}(\mathbf{s}_i).$$

We are also given the probabilities of generating $\mathbf{s}_1, \ldots, \mathbf{s}_N$, which we denote with $p(\mathbf{s}_i|\mathbf{x})$. Note that these probabilities can be obtained at no extra cost when generating sequences from the LLM. Our overall dataset is defined as

$$\mathcal{D} = (\mathbf{s}_1, \mathbf{m}_1, p(\mathbf{s}_1|\mathbf{x})), \ldots, (\mathbf{s}_N, \mathbf{m}_N, p(\mathbf{s}_N|\mathbf{x})).$$

The dataset can in general have repeated elements. It is important to note that we only have the probabilities for the sequences that were actually generated, which might represent a very small fraction of all possible sequences. Moreover, an important feature of our problem is that we want to achieve reasonable estimates using as few samples as possible. When generating the dataset, we do have the ability to ask for more data, i.e. increase $N$ until we are satisfied that our estimate of semantic entropy is good enough. We will make this precise in Section 3.

## 3    A Bayesian Estimator for Semantic Entropy

We now give a sketch of our estimation process. Since we have finite data, our estimate of semantic entropy will be noisy. Adopting the Bayesian philosophy, we construct a random variable $\mathbf{h}$ that represents our belief about the value of the semantic entropy $\mathrm{SE}_{\mathbf{x}}$, based on limited available data contained in a dataset $\mathcal{D}$ (we define $\mathbf{h}$ formally later on in the Section). Since we are motivated by detecting hallucinations, our focus is on measuring the quantities

$$\mathbb{E}\left[\mathbf{h}\right] \text{ and } \mathrm{Var}\left[\mathbf{h}\right], \tag{1}$$

i.e. the mean and variance of our Bayesian belief about what the value of the semantic entropy is.

In this Section, we describe a Bayesian process for forming a probabilistic belief over $\mathbf{h}$. For presentation purposes, we first derive our estimator under the assumption that that the number of meaning classes is known, i.e. $|M_{\mathbf{x}}| = K$. Under this assumption, in Section 3.1, we describe the basic variant of the estimator, which only uses the list of meanings $\mathbf{m}_1, \ldots, \mathbf{m}_N$. In Section 3.2, we then extend the estimator to also make use of the probabilities of the generated sequences $p(\mathbf{s}_1|\mathbf{x}), \ldots, p(\mathbf{s}_N|\mathbf{x})$. In Section 3.3, we remove the assumption that $K$ is known, defining a hierarchical Bayesian system that maintains a belief about $K$. In Section 3.4, we summarize our methodology in the form of pseudo-code.

### 3.1   Basic Variant of the Estimator

We first summarize the dataset, counting how many times we sampled each meaning. The counter for meaning $j \in M$ is defined as

$$\mathbf{c}_j = |\{i \ : \ \mathbf{m}_i = j\}|. \tag{2}$$

We have $\sum_j \mathbf{c}_j = N$. We seek to use the information from the counters to get an idea about how the true distribution over meanings looks like. We adopt the Bayesian modeling philosophy, using a belief distribution. Specifically, our Bayesian belief about the probability distribution over meanings is modeled as

$$B_p = \mathrm{Dirichlet}(\alpha + \mathbf{c}_1, \ldots, \alpha + \mathbf{c}_K), \tag{3}$$

where we used the letter $B_p$ to indicate that the probability distribution is used as a belief and $K$ is the number of meanings. The value $\alpha$ represents a prior of the Dirichlet distribution and is a hyper-parameter

of our method[6]. Equation 3 has a Bayesian interpretation as the posterior distribution, when the prior is chosen to be Dirichlet$(\alpha, , \ldots, \alpha)$, and the likelihood is categorical. Consider a random variable distributed according to $B_p$:

$$\mathbf{b} \sim B_p.$$

Here $\mathbf{b} \in \Delta^K$ is itself a probability distribution, representing the fraction of the total probability mass assigned to each meaning. A belief about $\mathbf{b}$ induces a belief about its entropy, represented with the random variable

$$\mathbf{h} \triangleq \mathbb{H}[\mathbf{b}].$$

The expectation as per equation 1 can be computed analytically as

$$\mathbb{E}\left[\mathbf{h}\right] = \int_{\mathbf{b}} \mathbb{H}[\mathbf{b}] p_{B_p}(\mathbf{b}) d\mathbf{b} = \psi\left(1 + K\alpha + \sum_j \mathbf{c}_j\right) - \sum_j \frac{\alpha + \mathbf{c}_j}{K\alpha + \sum_j' \mathbf{c}_j'} \psi\left(\alpha + \mathbf{c}_j + 1\right),$$

where $\psi$ is the digamma function (see Appendix A.1 for proof and a derivation of a similar expression for the variance integral).

## 3.2 An Estimator that Also Uses Sequence Probabilities

An LLM doesn't just give us meaning-classes but also probabilities of the generated continuations. Conditioning on this additional information can make our estimates of semantic entropy much better. Recall that the probability of generating $\mathbf{s}$ is denoted with $p(\mathbf{s}_i|\mathbf{x})$. We can define a constraint bounding the probability of each meaning, writing

$$\text{constr}(\mathbf{b}, \mathcal{D}) \coloneqq \left\{\mathbf{b}_j \geq \sum_{\mathbf{s} \in \{\mathbf{s} \,:\, \mathbf{s} \in \mathcal{D} \,,\, f^{\mathbf{x}}(\mathbf{s}) = j\}} p(\mathbf{s}|\mathbf{x})\right\}_{j=1,\ldots,K}. \tag{4}$$

Intuitively, equation 4 holds because the probability of a meaning $j$ is at least equal to the sum of probabilities of distinct sequences with that meaning. The bound is not an equality because it is possible (and typically the case) that we didn't generate all sequences that correspond to this meaning. Probabilistically, the constraint can be interpreted as an event, i.e. something we can condition on. We in fact do that, modifying the estimator to sample from belief conditional on the constraint:

$$\mathbf{b} \sim B_p \mid \text{constr}.$$

This conditioning is in fact key to obtaining good empirical results and is, as far as we can tell, novel to our approach. While the conditioning operation allows us to leverage all information at our disposal, it also makes the process of computing the expectation in equation 1 more complicated. In practice, the integrals for $\mathbb{E}[\mathbf{h}]$ and $\text{Var}[\mathbf{h}]$, which are defined as:

$$\mathbb{E}\left[\mathbf{h}\right] = \int_{\mathbf{b}} \mathbb{H}[\mathbf{b}] p_B^{\text{trunc}}(\mathbf{b}; \mathcal{D}) d\mathbf{b},$$

$$\text{Var}\left[\mathbf{h}\right] = \left(\int_{\mathbf{b}} \mathbb{H}[\mathbf{b}]^2 p_B^{\text{trunc}}(\mathbf{b}; \mathcal{D}) d\mathbf{b}\right) - \left(\mathbb{E}\left[\mathbf{h}\right]\right)^2,$$

will have to be computed approximately using a Monte Carlo method. Here, the density of a truncated Dirichlet random variable is defined as

$$p_B^{\text{trunc}}(\mathbf{b}; \mathcal{D}) = \begin{cases} \dfrac{p_B(\mathbf{b})}{\int_{\mathbf{b} \in \text{constr}(\mathbf{b}, \mathcal{D})} p_B(\mathbf{b}) d\mathbf{b}}, & \text{if } \mathbf{b} \in \text{constr}(\mathbf{b}, \mathcal{D}), \\ 0 & \text{if } \mathbf{b} \notin \text{constr}(\mathbf{b}, \mathcal{D}), \end{cases} \tag{5}$$

---

[6]We study the sensitivity of our method to the choice of $\alpha$ in Appendix D.

where we used the notation $p_B(\mathbf{b})$ to denote the Dirichlet PDF. We treat a particular choice of the integration algorithm as an implementation detail and defer its discussion to Appendix A.2. Note that, even though we are using a Monte Carlo method, obtaining estimates of semantic entropy is is still relatively cheap. This is because the integration routine is orders of magnitude cheaper than increasing $N$. In other words, sampling meanings from an LLM is expensive while MC integration has negligible cost.

### 3.3 Unknown Number of Meanings

Previously, we assumed that we know the number of meanings possible for a given context $x$, i.e. $|M_x| = K$ for a known value of $K$ that can be used to design the estimator. This is not a realistic assumption since the number of possible meanings can be vastly different for each context and is not typically known a priori. We resolve this dilemma in a Bayesian way, representing our Bayesian belief about the number of meanings using a probability distribution.

$$B_K = \text{Discrete}((K_1, \lambda_1), \dots, (K_M, \lambda_M)). \tag{6}$$

Here, the parameters $\lambda_1, \dots, \lambda_M$ are relative frequencies of each support size $K_i$. The parameters can be computed using a small (separate) training dataset and $M$ is the maximum observed support size.

Our belief about the number of meanings can be used to estimate the entropy in a hierarchical way. Specifically, given we have observed that there are at least $K_{\min}$ different meanings, we obtain the probability distribution

$$\mathbf{K} \sim B_K \mid (\mathbf{K} > K_{\min}).$$

Here, we used bold font for $\mathbf{K}$ to denote it is a random variable. We can use probabilities of this discrete distribution to take an expectation of entropy estimates conditioned on particular values of $K$:

$$\mathbb{E}[\mathbf{h}] = \mathbb{E}_{\mathbf{K}}\left[\mathbb{E}\left[\mathbf{h}|\mathbf{K}\right]\right], \quad \text{Var}[\mathbf{h}] = \mathbb{E}_{\mathbf{K}}\left[\text{Var}\left[\mathbf{h}|\mathbf{K}\right]\right] + \text{Var}_{\mathbf{K}}\left[\mathbb{E}\left[\mathbf{h}|\mathbf{K}\right]\right], \tag{7}$$

where the quantities $\mathbb{E}\left[\mathbf{h}|\mathbf{K}\right]$ and $\text{Var}\left[\mathbf{h}|\mathbf{K}\right]$ can be obtained as described in Section 3.2. In our pseudo-code (see the next Section), equation 7 is assumed to be implemented using a procedure AGGREGATESUPPORT.

### 3.4 Algorithm

**Stopping Rule**   Having specified the estimator for the quantities $\mathbb{E}[\mathbf{h}]$ and $\text{Var}[\mathbf{h}]$, we still need to specify the stopping rule for determining the right number of samples $N$. It is natural to keep drawing more samples until

$$\text{Var}[\mathbf{h}] \geq \gamma, \tag{8}$$

where $\gamma$ is a desired level of precision. Increasing $\gamma$ indicates we are satisfied with lower-quality semantic entropy estimates, which allows us to use smaller $N$. Intuitively, this stopping rule reflects the fact that we only want to know the semantic entropy to a certain level of precision. We summarize the ideas introduced in Sections 3.1, 3.2 and 3.3 in Algorithm 1.

**Benefits of the Bayesian Approach**   We will see in Section 6 that our Bayesian estimator works better than other semantic entropy estimators in practice. We now discuss the three theoretical reasons why this is the case. First, our estimator leverages a small training set to learn the prior, as in equation 6. Other estimators don't do this because, as they are not Bayesian, they do not have the concept of the prior.[7] Second, the Bayesian estimator combines all evidence (the probabilities of generated sequences and the meanings) in an optimal way, based on the Bayes rule. Other estimators don't do this. Third, our methodology gives us a handle on $\text{Var}[\mathbf{h}]$ (the variance in the belief about what the true semantic entropy is) allowing us to further improve sample efficiency by using stopping rule in equation 8.

---

[7]Note that the size of the training set does not scale with the number of queries to the estimator at inference time, hence the cost becomes entirely amortized.

---

**Algorithm 1** Estimate of Semantic Entropy for a prompt $\mathbf{x}$.

---
1:  $\mathcal{D} \leftarrow [\,]$
2: **repeat**
3:     $\mathbf{s}, p(\mathbf{s}|\mathbf{x}) \leftarrow \text{LLMSAMPLE}()$                               $\triangleright$ Sample $\mathbf{s}$ and store the corresponding probability.
4:     $\mathbf{m} \leftarrow f^{\mathbf{x}}(\mathbf{s})$                                        $\triangleright$ Determine the meaning.
5:     $\mathcal{D}.\text{append}(\mathbf{s}, \mathbf{m}, p(\mathbf{s}|\mathbf{x}))$
6:     $K_{\min} \leftarrow |\{\mathbf{m} \in \mathcal{D}\}|$
7:     **for** $j \in \{1, \ldots, K_{\min}\}$ **do**
8:         $\mathbf{c}_j \leftarrow |\{i \,:\, \mathbf{m}_i = j, \ \mathbf{m}_i \in \mathcal{D}\}|$                          $\triangleright$ Use equation 2.
9:     **end for**
10:     **for** $K \in \text{Support}\,(B_K \mid (\mathbf{K} > K_{\min}))$ **do**
11:         $p_B^{\text{trunc}}(\mathbf{b}; \mathcal{D}, K)$ is defined as per equation 5.          $\triangleright$ A truncated Dirichlet PDF.
12:         $\widehat{e}_K \leftarrow \text{NUMERICALLYINTEGRATE}(\int_{\mathbf{b}} \mathbb{H}[\mathbf{b}] p_B^{\text{trunc}}(\mathbf{b}; \mathcal{D}; K) d\mathbf{b})$
13:         $\widehat{e^2}_K \leftarrow \text{NUMERICALLYINTEGRATE}(\int_{\mathbf{b}} \mathbb{H}[\mathbf{b}]^2 p_B^{\text{trunc}}(\mathbf{b}; \mathcal{D}; K) d\mathbf{b})$
14:         $\widehat{var}_K = \widehat{e^2} - (\widehat{e})^2$
15:     **end for**
16:     $\widehat{e}, \widehat{var} \leftarrow \text{AGGREGATESUPPORT}(\widehat{e}_k, \widehat{var}_k)$          $\triangleright$ Apply equation 7 for $k \in \{K_{\min}, \ldots, K_{\max}\}$.
17: **until** $\widehat{var} \geq \gamma$
18: **return** $\widehat{e}$

---

## 4 Baselines

### 4.1 Other Estimators For Semantic Entropy

There are two existing baselines for measuring semantic entropy, both of which coming from the paper by Farquhar et al. (2024). They are called *discrete semantic entropy* and *semantic entropy* in that paper, although we use the term *histogram semantic entropy* for the former and *rescaled semantic entropy* for the latter to avoid confusion with the concept of semantic entropy itself, which is independent of the estimator used.

**Histogram Semantic Entropy** This estimator samples a fixed number of sequences, computes the meaning of each sequence and then computes the entropy of the empirical histogram of the meaning distribution. It is computed from the meaning counts $\mathbf{c}_j$ as $-\sum_j (\frac{\mathbf{c}_j}{N}) \log \frac{\mathbf{c}_j}{N}$.

**Rescaled Semantic Entropy** This estimator also samples a fixed number of sequences and then computes the meaning of each. However, it assigns probabilities to each meaning in a different way. First, one defines the un-normalized probability distribution

$$q(\mathbf{m}|\mathbf{x}) = \sum_{\mathbf{s} \in \{\mathbf{s}\,:\,\mathbf{s} \in \mathcal{D}\,,\,f^{\mathbf{x}}(\mathbf{s}) = \mathbf{m}\}} p(\mathbf{s}|\mathbf{x}).$$

This is then normalized as

$$p(\mathbf{m}|\mathbf{x}) = \frac{q(\mathbf{m}|\mathbf{x})}{\sum_{\mathbf{m}} q(\mathbf{m}|\mathbf{x})},$$

and the semantic entropy is computed as the Shannon entropy of this distribution. In addition, the probabilities $p(\mathbf{s}|\mathbf{x})$, which normally correspond to the multiplication of the probabilities of each token conditioned on past tokens, are heuristically replaced by the exponent of the mean log probability of each token, a process known as length normalization. We report results for the rescaled estimator both with and without the (heuristic) length normalization.

### 4.2 Other Baselines for Hallucination Detection

Semantic Entropy is not the only way of detecting hallucinations. We consider two non-entropy baselines.

**P(True)** It has been shown that LLMs have surprising introspective abilities, i.e. one can often see if the LLM is hallucinating simply by simply asking it (Kadavath et al., 2022). We use the same procedure for measuring P(True) as was employed by Farquhar et al. (2024).

**Sequence Log Likelihood** Recently, Aichberger et al. (2024) suggested using the log probability of the sequence generated greedily as a predictor of hallucinations, justifying it using the notion of zero-one scoring rule (Hofman et al., 2024). Unfortunately, this method is not directly compatible with the evaluation protocol of Farquhar et al. (2024), which does not perform greedy generation. In order to stay within the boundaries of that protocol, we instead used the log likelihood of a sequence generated with a low temperature.[8]

## 5 Prior Work

**Hallucination Detection** We do not provide a complete survey of hallucinations in Large Language Models, instead referring the reader to the work of Ji et al. (2023). We focus our work on combating 'confabulatory' hallucinations also addressed by Farquhar et al. (2024), i.e. situations where the LLM is randomly adding spurious facts to its replies. This is the same kind of hallucinations that was considered by Filippova (2020) and Maynez et al. (2020).

**Semantic Entropy** We build on the works that pioneered semantic entropy (Kuhn et al., 2023; Farquhar et al., 2024) by providing a more statistically efficient estimator. Our work is different from entropy probes (Kossen et al., 2024), which attempt to distill a thresholded version of semantic entropy into a classifier, although the ideas can certainly be used in conjunction with each other (similar ideas were also explored by Chen et al. (2024)). Our estimators do not attempt to leverage similarity in the meaning clusters (Nikitin et al., 2024; Qiu & Miikkulainen, 2024), instead focusing on obtaining as accurate estimates of vanilla semantic entropy for a given sample budget as possible.

**Bayesian Entropy Estimation** Wolpert & Wolf (1994) have provided the foundations for Bayesian estimators of entropy for arbitrary priors, and also pioneered the specialization to the case of the Dirichlet prior. Hausser & Strimmer (2009) have provided an explicit summary of the equivalences between the Dirichlet-Bayesian estimator and various pre-existing entropy estimators, for different values of the Dirichlet prior parameter. Archer et al. (2014) have provided an overview of past work on Bayesian entropy estimation, in addition to extending the framework to distributions with countably infinite support.[9]

**Epistemic Uncertainty in LLMs** The distinction between epistemic and aleatoric uncertainty (Gal et al., 2016; 2017; Kendall & Gal, 2017) has been proposed as a useful idea in modeling the behavior of LLMs (Abbasi Yadkori et al., 2024). In this paper, we do not distinguish between aleatoric and epistemic uncertainty, instead staying in the framework of Farquhar et al. (2024) and modeling the combined predictive uncertainty. While an accurate model of epistemic uncertainty would almost certainly lead to improved hallucination detection, we leave such extensions to further work.

**Human Perception of Hallucinations** Hallucinations are related to how confident LLMs are about their outputs. Recent research (Steyvers et al., 2025) studies how such self-confidence intrinsic in LLMs relates to how humans perceive it. Our work is largely orthogonal to this effort. Indeed, we treat the definition of semantic entropy as a given and focus on finding the statistically most efficient way to estimate it.

## 6 Experiments

### 6.1 Experimental Setup

**Evaluation Methodology** Our goal is to measure the quality of semantic entropy estimates as quantified with AUROC on hallucination detection tasks. To do so, we follow the methodology from the paper by

---

[8]The temperature was set to 0.1.

[9]We do not use their infinite support framework, instead modeling unknown support using techniques described in Section 3.3.

$N = 2$

| LLM | Dataset | LL | P(true) | SE-Bayes | SE-Histogram | SE-Rescaled | SE-Rescaled (h) |
|---|---|---|---|---|---|---|---|
| | NQ | $0.583 \pm 0.000$ | $0.461 \pm 0.000$ | $\mathbf{0.723 \pm 0.007}$ | $0.652 \pm 0.010$ | $0.654 \pm 0.013$ | $0.644 \pm 0.014$ |
| Llama-2 | SVAMP | $0.631 \pm 0.000$ | $0.469 \pm 0.000$ | $\mathbf{0.855 \pm 0.022}$ | $0.748 \pm 0.024$ | $0.749 \pm 0.027$ | $0.759 \pm 0.025$ |
| | Squad | $0.624 \pm 0.000$ | $0.441 \pm 0.000$ | $\mathbf{0.735 \pm 0.007}$ | $0.654 \pm 0.012$ | $0.659 \pm 0.017$ | $0.649 \pm 0.010$ |
| | Trivia QA | $0.594 \pm 0.000$ | $0.436 \pm 0.000$ | $\mathbf{0.737 \pm 0.006}$ | $0.670 \pm 0.012$ | $0.675 \pm 0.012$ | $0.673 \pm 0.010$ |
| | NQ | $0.627 \pm 0.000$ | $0.615 \pm 0.000$ | $\mathbf{0.728 \pm 0.008}$ | $0.640 \pm 0.013$ | $0.663 \pm 0.017$ | $0.649 \pm 0.020$ |
| Llama-3.2 | SVAMP | $0.647 \pm 0.000$ | $0.414 \pm 0.000$ | $\mathbf{0.845 \pm 0.022}$ | $0.761 \pm 0.032$ | $0.774 \pm 0.030$ | $0.771 \pm 0.028$ |
| | Squad | $0.610 \pm 0.000$ | $0.555 \pm 0.000$ | $\mathbf{0.664 \pm 0.019}$ | $0.608 \pm 0.024$ | $\mathbf{0.642 \pm 0.029}$ | $\mathbf{0.621 \pm 0.027}$ |
| | Trivia QA | $0.614 \pm 0.000$ | $0.710 \pm 0.000$ | $\mathbf{0.768 \pm 0.004}$ | $0.699 \pm 0.005$ | $0.706 \pm 0.007$ | $0.708 \pm 0.008$ |
| Llama-3.3-70B | Trivia QA | $0.556 \pm 0.000$ | $0.686 \pm 0.000$ | $\mathbf{0.775 \pm 0.003}$ | $0.653 \pm 0.006$ | $0.650 \pm 0.006$ | $0.651 \pm 0.006$ |
| | NQ | $0.695 \pm 0.000$ | $\mathbf{0.731 \pm 0.000}$ | $0.705 \pm 0.013$ | $0.647 \pm 0.007$ | $0.700 \pm 0.007$ | $0.654 \pm 0.009$ |
| Mistral | SVAMP | $0.645 \pm 0.000$ | $0.843 \pm 0.000$ | $\mathbf{0.876 \pm 0.011}$ | $0.793 \pm 0.023$ | $0.815 \pm 0.028$ | $0.812 \pm 0.024$ |
| | Squad | $\mathbf{0.698 \pm 0.000}$ | $0.687 \pm 0.000$ | $0.667 \pm 0.010$ | $0.618 \pm 0.005$ | $0.671 \pm 0.017$ | $0.631 \pm 0.010$ |
| | Trivia QA | $0.672 \pm 0.000$ | $0.647 \pm 0.000$ | $\mathbf{0.682 \pm 0.008}$ | $0.638 \pm 0.011$ | $0.645 \pm 0.013$ | $0.643 \pm 0.012$ |

$N = 5$

| LLM | Dataset | LL | P(true) | SE-Bayes | SE-Histogram | SE-Rescaled | SE-Rescaled (h) |
|---|---|---|---|---|---|---|---|
| | NQ | $0.583 \pm 0.000$ | $0.461 \pm 0.000$ | $\mathbf{0.752 \pm 0.006}$ | $0.730 \pm 0.009$ | $0.701 \pm 0.012$ | $0.725 \pm 0.010$ |
| Llama-2 | SVAMP | $0.631 \pm 0.000$ | $0.469 \pm 0.000$ | $\mathbf{0.871 \pm 0.011}$ | $\mathbf{0.849 \pm 0.012}$ | $\mathbf{0.853 \pm 0.013}$ | $\mathbf{0.858 \pm 0.013}$ |
| | Squad | $0.624 \pm 0.000$ | $0.441 \pm 0.000$ | $\mathbf{0.774 \pm 0.008}$ | $0.756 \pm 0.004$ | $0.711 \pm 0.012$ | $0.752 \pm 0.005$ |
| | Trivia QA | $0.594 \pm 0.000$ | $0.436 \pm 0.000$ | $\mathbf{0.763 \pm 0.009}$ | $0.734 \pm 0.007$ | $0.737 \pm 0.006$ | $0.735 \pm 0.006$ |
| | NQ | $0.627 \pm 0.000$ | $0.615 \pm 0.000$ | $\mathbf{0.760 \pm 0.008}$ | $0.732 \pm 0.012$ | $0.691 \pm 0.005$ | $0.733 \pm 0.012$ |
| Llama-3.2 | SVAMP | $0.647 \pm 0.000$ | $0.414 \pm 0.000$ | $\mathbf{0.870 \pm 0.009}$ | $0.860 \pm 0.010$ | $0.850 \pm 0.004$ | $\mathbf{0.864 \pm 0.009}$ |
| | Squad | $0.610 \pm 0.000$ | $0.555 \pm 0.000$ | $\mathbf{0.710 \pm 0.017}$ | $\mathbf{0.707 \pm 0.012}$ | $0.667 \pm 0.021$ | $\mathbf{0.705 \pm 0.013}$ |
| | Trivia QA | $0.614 \pm 0.000$ | $0.710 \pm 0.000$ | $\mathbf{0.792 \pm 0.004}$ | $0.775 \pm 0.002$ | $0.763 \pm 0.005$ | $0.777 \pm 0.004$ |
| Llama-3.3-70B | Trivia QA | $0.556 \pm 0.000$ | $0.686 \pm 0.000$ | $\mathbf{0.793 \pm 0.005}$ | $0.750 \pm 0.005$ | $0.740 \pm 0.004$ | $0.747 \pm 0.006$ |
| | NQ | $0.695 \pm 0.000$ | $0.731 \pm 0.000$ | $\mathbf{0.780 \pm 0.006}$ | $0.762 \pm 0.007$ | $0.728 \pm 0.008$ | $0.756 \pm 0.007$ |
| Mistral | SVAMP | $0.645 \pm 0.000$ | $0.843 \pm 0.000$ | $\mathbf{0.880 \pm 0.019}$ | $\mathbf{0.866 \pm 0.021}$ | $0.855 \pm 0.027$ | $\mathbf{0.878 \pm 0.022}$ |
| | Squad | $0.698 \pm 0.000$ | $0.687 \pm 0.000$ | $\mathbf{0.731 \pm 0.008}$ | $\mathbf{0.719 \pm 0.010}$ | $0.699 \pm 0.008$ | $0.712 \pm 0.008$ |
| | Trivia QA | $0.672 \pm 0.000$ | $0.647 \pm 0.000$ | $\mathbf{0.691 \pm 0.005}$ | $\mathbf{0.688 \pm 0.005}$ | $0.684 \pm 0.004$ | $\mathbf{0.690 \pm 0.005}$ |

Table 1: Measured AUROC for a fixed budget of $N = 2$ and $N = 5$ samples per prompt.

Farquhar et al. (2024) as much as possible, deviating from it only by (1) separating out the dataset generation phase and the entropy estimation phase, (2) varying the sample budget $N$ and (3) removing bugs from the dataset generation code. We defer the detailed discussion of the methodology to Appendix B.

**LLMs and Source Datasets** We investigate the behavior of four LLMs. We use Llama-2-7b-chat for comparability with Farquhar et al. (2024). We also evaluate on the much more modern Llama-3.2-3B-Instruct, the large Llama-3.3-70B-Instruct and on Mistral-Small-24B-Instruct-2501. These LLMs are referred to as Llama 2, Llama 3, Llama-3.3-70 and Mistral in our figures. We used the TriviaQA (Joshi et al., 2017), SQUAD (Rajpurkar et al., 2016), SVAMP (Patel et al., 2021) and NQ (Lee et al., 2019) datasets. Due to computational constraints, we only ran the largest LLM on one dataset (TriviaQA).

**Derivative Entropy Estimation Datasets** For each combination of LLM and dataset, we generated a derivative dataset of 100 LLM generations per prompt for 1000 prompts, which we then used to estimate semantic entropy. We will release these derivative datasets upon acceptance allowing researches without GPU access to work on even better estimators for semantic entropy.

**Train and Test** Our Bayesian Semantic Entropy estimator requires a training set to estimate the prior on the size of the support of the meaning distribution as in equation 6. We use the first 200 prompts from each derivative dataset as the training set and the remaining 800 as the test set.

**Temperature**    Following the methodology of Farquhar et al. (2024), the $N$ LLM responses are generated with temperature 1.0. On the other hand, the LLM response about which we seek to determine if it is a hallucination is generated with temperature 0.1. Because GPU response generation is expensive, we did not tune those temperatures.

## 6.2    Results

We performed two types of experiment. First, we studied the setting of a fixed budget of samples per prompt. Second, we varied the number of samples per prompt, giving harder prompts more data. In both cases, we note that we can support $N = 1$ because we have access to the probability of the generated sequence, which already gives us an imperfect but still useful handle on the entropy (for example, if the probability is close to one, we know that the entropy is almost zero).

**Fixed Budget Per Prompt**    Results for $N = 2$ and $N = 5$ samples per prompt[10] are shown in Table 1. Bold font is applied as follows: the estimator with the best mean performance is put in bold, together with all the others with overlapping confidence bars. It can be seen that the Bayesian estimator mostly outperformed or tied with other approaches to measuring semantic entropy, with the difference being greater for small $N$. We can also see that it is difficult to conclude which version of the rescaled estimator is better.

**Main Experiment: Adaptive Budget Per Prompt**    As described in Section 3.4, the Bayesian framework gives us an additional handle on sample complexity in that we can use the variance of the belief about the semantic entropy as a proxy for confidence. Results are shown in Figures 1, 2, 3 and 4. All confidence bars for the AUROC estimates in our paper represent 1.96 times the standard error. It can be seen that our Bayesian estimator is nearly Pareto-optimal in the sense that we achieve better AUROC than other approaches to semantic entropy, regardless of the value of $N$. Note that performance of the adaptive Bayesian estimator for a given $N$ will in general be better than performance for the same fixed value of $N$. This is because, while the number of prompts is still $N$ on average, harder prompts will get more samples (and easier prompts will get fewer). Concerning the non-semantic-entropy baselines, we outperform them for all $N$ for Llama 2 and 3, while needing $N \geq 3$ for Mistral. We stress one additional take-away from the experiment: our Bayesian estimator is often competitive even for $N = 1$. This is completely counterintuitive since the entailment oracle (a crucial component of semantic entropy) is not needed in that case.

**Note on Consistency**    By definition, for a large enough sample budget, any consistent estimator will produce the same correct value of semantic entropy, which will give rise to the same value of AUROC. There are two reasons why such convergence does not happen in practice in our experiments. First, not all compared estimators are consistent (for example the rescaled estimator with heuristic sequence probabilities is not). Second, convergence to the exact same value might require a sample budged far in excess of 10.

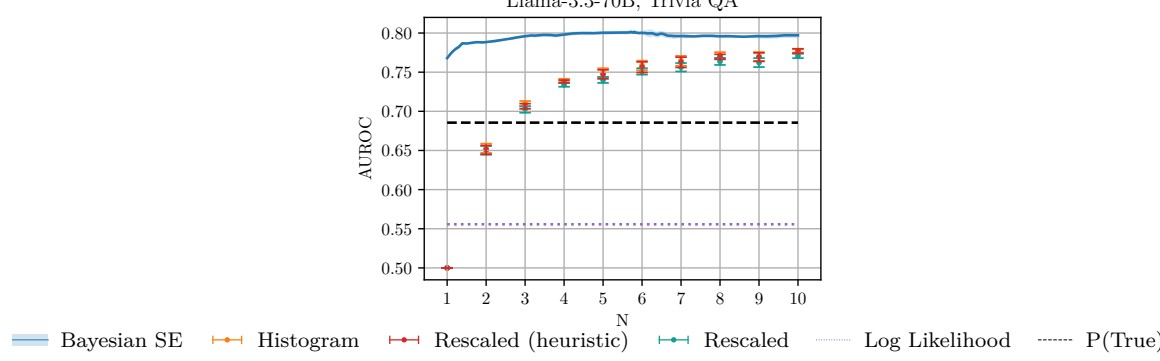

Figure 1: Results in the adaptive budget setting (Llama 3.3, 70 billion parameters).

---

[10]See Appendix C for results for other values of $N$.

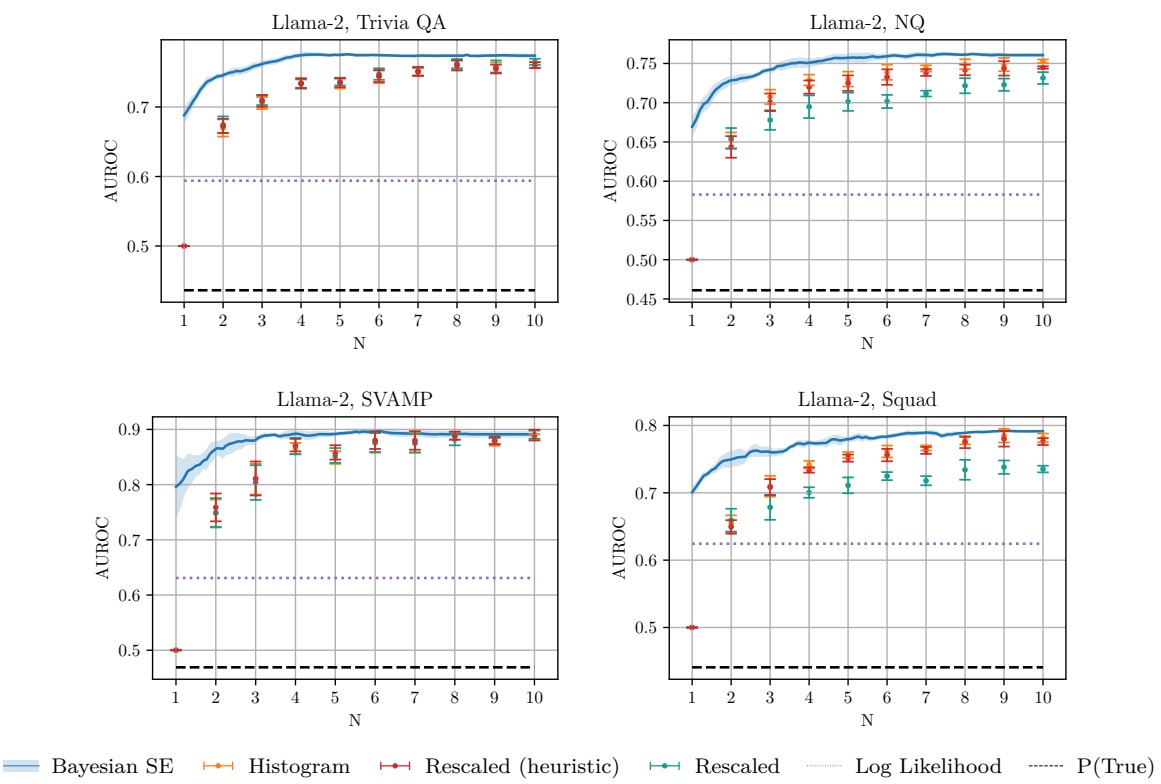

Figure 2: Results in the adaptive budget setting (Llama 2).

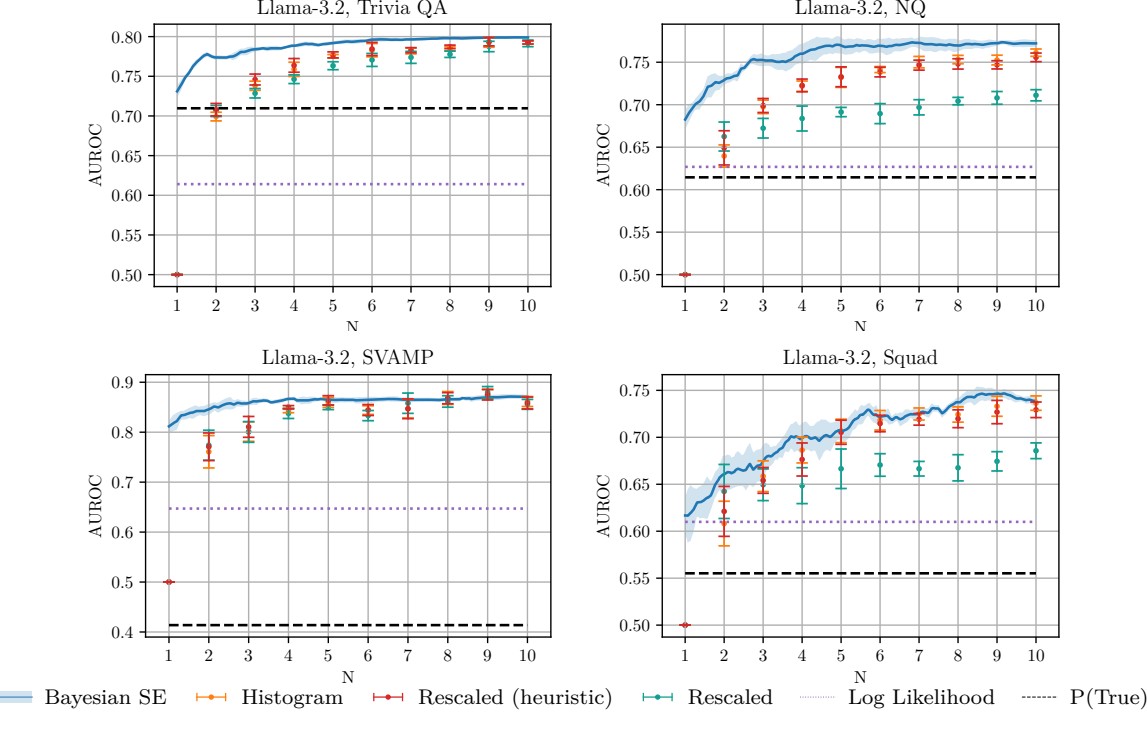

Figure 3: Results in the adaptive budget setting (Llama 3).

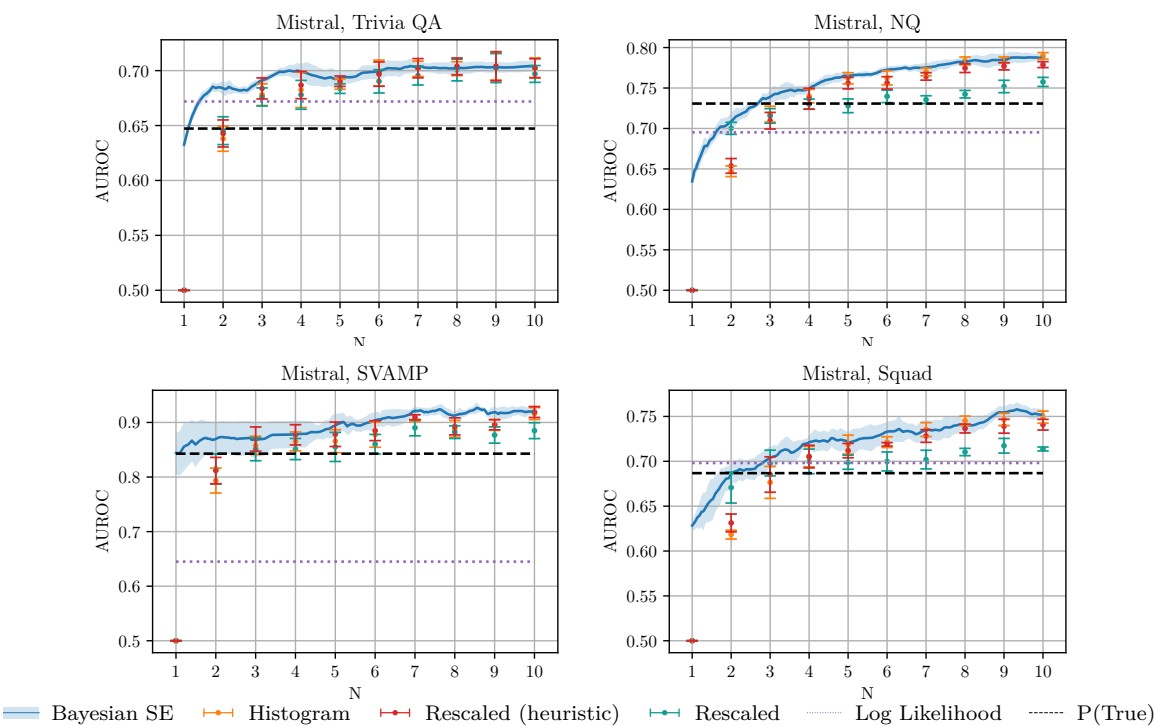

Figure 4: Results in the adaptive budget setting (Mistral).

# 7    Conclusions, Limitations and Perspectives

We have described a new Bayesian estimator for measuring semantic entropy. The proposed estimator has systematically outperformed other semantic entropy baselines in several practical settings. Like any Bayesian method, our approach is constrained by the choice of the prior, where our choice can be viewed as limited in expressivity. This can be mitigated by using more complex hierarchical priors, for example along the lines proposed by Nemenman et al. (2001). We leave this to further work.

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

## A  Integral Computation

### A.1  Mean and Variance of Expected Entropy under the Dirichlet Distribution

Here, we derive the analytical expressions for the mean $\mathbb{E}[\mathbf{h}]$ and the second moment $\mathbb{E}[\mathbf{h}^2]$ of the entropy

$$\mathbf{h} \triangleq \mathbb{H}[\mathbf{b}] = -\sum_{i=1}^{K} b_i \log b_i,$$

where the probability vector $\mathbf{b} = (b_1, \ldots, b_K)$ follows a Dirichlet distribution $\mathbf{b} \sim \text{Dir}(\boldsymbol{\alpha})$ with parameters $\boldsymbol{\alpha} = (\alpha_1, \ldots, \alpha_K)$. Let $\alpha_0 = \sum_{i=1}^{K} \alpha_i$. The results involving digamma ($\psi$) and trigamma ($\psi_1$) functions are based on standard properties of the Dirichlet distribution (Wolpert & Wolf, 1994; Hausser & Strimmer, 2009).

**Mean Entropy $\mathbb{E}[\mathbf{h}]$**  Using the linearity of expectation and the known expectation $\mathbb{E}[b_i \log b_i]$ for a Dirichlet distribution:

$$
\begin{aligned}
\mathbb{E}[\mathbf{h}] &= \mathbb{E}\left[-\sum_{i=1}^{K} b_i \log b_i\right] = -\sum_{i=1}^{K} \mathbb{E}[b_i \log b_i] \\
&= -\sum_{i=1}^{K} \frac{\alpha_i}{\alpha_0} \left(\psi(\alpha_i + 1) - \psi(\alpha_0 + 1)\right) \\
&= \psi(\alpha_0 + 1) \sum_{i=1}^{K} \frac{\alpha_i}{\alpha_0} - \sum_{i=1}^{K} \frac{\alpha_i}{\alpha_0} \psi(\alpha_i + 1) \\
&= \psi(\alpha_0 + 1) - \sum_{i=1}^{K} \frac{\alpha_i}{\alpha_0} \psi(\alpha_i + 1)
\end{aligned}
$$

This formula corresponds to the one used to compute $\mathbb{E}[\mathbf{h}]$ in Section 3.1 (after substituting the appropriate parameters $\alpha + \mathbf{c}_j$).

**Second Moment $\mathbb{E}[\mathbf{h}^2]$**  We start by expanding the square of the entropy:

$$\mathbf{h}^2 = \left(-\sum_{i=1}^{K} b_i \log b_i\right)^2 = \sum_{i=1}^{K} \sum_{j=1}^{K} (b_i b_j \log b_i \log b_j)$$

Applying the expectation:

$$
\begin{aligned}
\mathbb{E}[\mathbf{h}^2] &= \mathbb{E}\left[\sum_{i=1}^{K} \sum_{j=1}^{K} (b_i b_j \log b_i \log b_j)\right] \\
&= \sum_{i=1}^{K} \sum_{j=1}^{K} \mathbb{E}[b_i b_j \log b_i \log b_j] \\
&= \sum_{i=1}^{K} \mathbb{E}[b_i^2 (\log b_i)^2] + \sum_{i \neq j} \mathbb{E}[b_i b_j \log b_i \log b_j]
\end{aligned}
$$

The calculation requires the expectations $\mathbb{E}[b_i^2 (\log b_i)^2]$ and $\mathbb{E}[b_i b_j \log b_i \log b_j]$ for $i \neq j$. Before the derivation of these quantities, let us derive some useful lemmas.

**Lemma A.1.** *Let $\mathbf{b} \sim Dir(\boldsymbol{\alpha})$, where $\boldsymbol{\alpha} = (\alpha_1, \ldots, \alpha_K)$. Let $i \in \{1, \ldots, K\}$. Then:*

$$\frac{\partial}{\partial \alpha_i} \log p(\mathbf{b}|\boldsymbol{\alpha}) = \psi(\alpha_0) - \psi(\alpha_i) + \log b_i,$$

*where $\psi(x)$ is the digamma function.*

*Proof.* First, let us re-write this quantity:

$$\frac{\partial}{\partial \alpha_i} \log p(\mathbf{b}|\boldsymbol{\alpha}) = \frac{\partial}{\partial \alpha_i} \left( \log \Gamma(\alpha_0) - \sum_{k=1}^{K} \log \Gamma(\alpha_k) + \sum_{k=1}^{K} (\alpha_k - 1) \log b_k \right)$$

Using the chain rule:

$$\frac{\partial}{\partial \alpha_i} \log \Gamma(\alpha_0) = \frac{d \log \Gamma(\alpha_0)}{d\alpha_0} \frac{\partial \alpha_0}{\partial \alpha_i} = \psi(\alpha_0) \cdot 1 = \psi(\alpha_0).$$

Therefore:

$$\frac{\partial}{\partial \alpha_i} \log p(\mathbf{b}|\boldsymbol{\alpha}) = \psi(\alpha_0) - \frac{d \log \Gamma(\alpha_i)}{d\alpha_i} + \log b_i$$
$$= \psi(\alpha_0) - \psi(\alpha_i) + \log b_i.$$

$\square$

**Lemma A.2.** *Let* $\mathbf{b} \sim Dir(\boldsymbol{\alpha})$*, where* $\boldsymbol{\alpha} = (\alpha_1, \ldots, \alpha_K)$*. Let* $i \in \{1, \ldots, K\}$*. Then:*

$$\mathbb{E}[(\log b_i)^2] = \big(\psi_1(\alpha_i) - \psi_1(\alpha_0)\big) + \big(\psi(\alpha_i) - \psi(\alpha_0)\big)^2,$$

*where* $\psi(x)$ *is the digamma function.*

*Proof.* It is known that:

$$\mathbb{E}[\log b_i] = \psi(\alpha_i) - \psi(\alpha_0).$$

It follows that:

$$\int p(\mathbf{b}|\boldsymbol{\alpha}) \left[\psi(\alpha_0) - \psi(\alpha_i) + \log b_i\right] d\mathbf{b} = \psi(\alpha_0) - \psi(\alpha_i) + \mathbb{E}[\log b_i] = 0.$$

If we apply a derivative to this quantity, it must be 0, since it is a constant:

$$\frac{\partial}{\partial \alpha_i} \int p(\mathbf{b}|\boldsymbol{\alpha}) \left[\psi(\alpha_0) - \psi(\alpha_i) + \log b_i\right] d\mathbf{b} = 0$$

For this integral, we can apply the Leibniz integral rule for differentiation under the integral sign. Applying the product rule for differentiation under the integral sign we get:

$$\int \left(\frac{\partial p(\mathbf{b}|\boldsymbol{\alpha})}{\partial \alpha_i}\right) \left[\psi(\alpha_0) - \psi(\alpha_i) + \log b_i\right] d\mathbf{b} + \int p(\mathbf{b}|\boldsymbol{\alpha}) \frac{\partial}{\partial \alpha_i} \left[\psi(\alpha_0) - \psi(\alpha_i) + \log b_i\right] d\mathbf{b} = 0$$

By using Lemma A.1 and the log-derivative trick, we can substitute $\frac{\partial p}{\partial \alpha_i} = p \frac{\partial \log p}{\partial \alpha_i} = p \cdot (\psi(\alpha_0) - \psi(\alpha_i) + \log b_i)$:

$$\int p(\mathbf{b}|\boldsymbol{\alpha}) \left(\psi(\alpha_0) - \psi(\alpha_i) + \log b_i\right)^2 d\mathbf{b} + \int p(\mathbf{b}|\boldsymbol{\alpha}) \left[\psi_1(\alpha_0) - \psi_1(\alpha_i)\right] d\mathbf{b} = 0,$$

where $\frac{\partial}{\partial \alpha_i} \psi(\alpha_0) = \psi_1(\alpha_0) \frac{\partial \alpha_0}{\partial \alpha_i} = \psi_1(\alpha_0)$, $\frac{\partial}{\partial \alpha_i} \psi(\alpha_i) = \psi_1(\alpha_i)$, and $\frac{\partial}{\partial \alpha_i} \log b_i = 0$.

We can further simplify this expression by using the expected value definition:

$$\underbrace{\mathbb{E}[(\psi(\alpha_0) - \psi(\alpha_i) + \log b_i)^2]}_{(1)} + \underbrace{\psi_1(\alpha_0) - \psi_1(\alpha_i)}_{(2)} = 0 \ .$$

Let $C_i = \psi(\alpha_0) - \psi(\alpha_i) = -\mathbb{E}[\log b_i]$. Term (1) becomes:

$$\mathbb{E}[(C_i + \log b_i)^2] = \mathbb{E}[C_i^2 + 2C_i \log b_i + (\log b_i)^2]$$

$$= C_i^2 + 2C_i \mathbb{E}[\log b_i] + \mathbb{E}[(\log b_i)^2]$$

Substitute $\mathbb{E}[\log b_i] = -C_i$:

$$= C_i^2 + 2C_i(-C_i) + \mathbb{E}[(\log b_i)^2] = -C_i^2 + \mathbb{E}[(\log b_i)^2]$$

Substituting this back into the equation derived from differentiation:

$$(-C_i^2 + \mathbb{E}[(\log b_i)^2]) + \psi_1(\alpha_0) - \psi_1(\alpha_i) = 0$$

$$\mathbb{E}[(\log b_i)^2] = C_i^2 - \psi_1(\alpha_0) + \psi_1(\alpha_i).$$

Substitute $C_i = \psi(\alpha_0) - \psi(\alpha_i)$:

$$\mathbb{E}[(\log b_i)^2] = (\psi(\alpha_0) - \psi(\alpha_i))^2 + \psi_1(\alpha_i) - \psi_1(\alpha_0).$$

Rearranging gives the desired result:

$$\mathbb{E}[(\log b_i)^2] = (\psi_1(\alpha_i) - \psi_1(\alpha_0)) + (\psi(\alpha_i) - \psi(\alpha_0))^2.$$

$\square$

**Lemma A.3.** *Let $\mathbf{b} \sim Dir(\boldsymbol{\alpha})$, where $\boldsymbol{\alpha} = (\alpha_1, \ldots, \alpha_K)$. Let $i, j \in \{1, \ldots, K\}$ and $i \neq j$. Then:*

$$\mathbb{E}[\log b_i \log b_j] = -\psi_1(\alpha_0) + (\psi(\alpha_i) - \psi(\alpha_0))(\psi(\alpha_j) - \psi(\alpha_0)),$$

*where $\psi(x)$ is the digamma function and $\psi_1(x) = \frac{d}{dx}\psi(x)$ is the trigamma function.*

*Proof.* From Lemma A.1, we know that, for a given $i \in \{1, \ldots, K\}$, the following quantity is 0:

$$\int p(\mathbf{b}|\boldsymbol{\alpha}) [\psi(\alpha_0) - \psi(\alpha_i) + \log b_i] \, d\mathbf{b} = 0.$$

Now, we differentiate this quantity with respect to $\alpha_j$, where $j \neq i$:

$$\frac{\partial}{\partial \alpha_j} \int p(\mathbf{b}|\boldsymbol{\alpha}) [\psi(\alpha_0) - \psi(\alpha_i) + \log b_i] \, d\mathbf{b} = 0.$$

For this integral, we can apply the Leibniz integral rule for differentiation under the integral sign. Applying the product rule for differentiation under the integral sign we get:

$$\int \left(\frac{\partial p(\mathbf{b}|\boldsymbol{\alpha})}{\partial \alpha_j}\right) [\psi(\alpha_0) - \psi(\alpha_i) + \log b_i] \, d\mathbf{b} + \int p(\mathbf{b}|\boldsymbol{\alpha}) \frac{\partial}{\partial \alpha_j} [\psi(\alpha_0) - \psi(\alpha_i) + \log b_i] \, d\mathbf{b} = 0$$

Due to Lemma A.1 and the log-derivative trick, we can substitute $\frac{\partial p}{\partial \alpha_j} = p \frac{\partial \log p}{\partial \alpha_j} = p \cdot (\psi(\alpha_0) - \psi(\alpha_j) + \log b_j)$:

$$\int p(\mathbf{b}|\boldsymbol{\alpha}) (\psi(\alpha_0) - \psi(\alpha_j) + \log b_j)(\psi(\alpha_0) - \psi(\alpha_i) + \log b_i) \, d\mathbf{b} + \int p(\mathbf{b}|\boldsymbol{\alpha}) [\psi_1(\alpha_0)] \, d\mathbf{b} = 0,$$

where $\frac{\partial}{\partial \alpha_j}\psi(\alpha_0) = \psi_1(\alpha_0)$, $\frac{\partial}{\partial \alpha_j}\psi(\alpha_i) = 0$ since $j \neq i$, and $\frac{\partial}{\partial \alpha_j}\log b_i = 0$.

We can re-write the previous quantity by using the expected value definition:

$$\underbrace{\mathbb{E}[(\psi(\alpha_0) - \psi(\alpha_j) + \log b_j)(\psi(\alpha_0) - \psi(\alpha_i) + \log b_i)]}_{(1)} + \underbrace{\psi_1(\alpha_0)}_{(2)} = 0$$

Let $C_i = \psi(\alpha_0) - \psi(\alpha_i) = -\mathbb{E}[\log b_i]$ and $C_j = \psi(\alpha_0) - \psi(\alpha_j) = -\mathbb{E}[\log b_j]$. Term (1) becomes:

$$\mathbb{E}[(C_j + \log b_j)(C_i + \log b_i)] = \mathbb{E}[C_i C_j + C_i \log b_j + C_j \log b_i + \log b_i \log b_j]$$
$$= C_i C_j + C_i \mathbb{E}[\log b_j] + C_j \mathbb{E}[\log b_i] + \mathbb{E}[\log b_i \log b_j]$$
$$= C_i C_j + C_i(-C_j) + C_j(-C_i) + \mathbb{E}[\log b_i \log b_j]$$
$$= C_i C_j - C_i C_j - C_j C_i + \mathbb{E}[\log b_i \log b_j] = -C_i C_j + \mathbb{E}[\log b_i \log b_j].$$

Substituting this back into the equation derived from differentiation:

$$(-C_i C_j + \mathbb{E}[\log b_i \log b_j]) + \psi_1(\alpha_0) = 0$$

$$\mathbb{E}[\log b_i \log b_j] = C_i C_j - \psi_1(\alpha_0)$$

Substitute $C_i = \psi(\alpha_0) - \psi(\alpha_i)$ and $C_j = \psi(\alpha_0) - \psi(\alpha_j)$:

$$\mathbb{E}[\log b_i \log b_j] = (\psi(\alpha_0) - \psi(\alpha_i))(\psi(\alpha_0) - \psi(\alpha_j)) - \psi_1(\alpha_0).$$

Rearranging gives the desired result:

$$\mathbb{E}[\log b_i \log b_j] = -\psi_1(\alpha_0) + (\psi(\alpha_i) - \psi(\alpha_0))(\psi(\alpha_j) - \psi(\alpha_0)) \quad \text{for } i \neq j.$$

$\square$

**Lemma A.4.** *For $\mathbf{b} \sim Dir(\boldsymbol{\alpha})$,*

$$\mathbb{E}[b_{i_1} \ldots b_{i_n} f(\mathbf{b})] = \mathbb{E}[b_{i_1} \ldots b_{i_n}] \cdot \mathbb{E}'[f(\mathbf{b}')]$$

*where $i_1, \ldots, i_n \in \{1, \ldots, K\}$ are $n$ indices, $\mathbf{b}' \sim Dir(\boldsymbol{\alpha} + \sum_{k=1}^n \mathbf{e}_{i_k})$, and $\mathbf{e}_k$ is the $k$-th standard basis vector.*

*Proof.* First, let us re-write the term $b_{i_1} \ldots b_{i_n}$ on the counts $c_k$:

$$b_{i_1} \ldots b_{i_n} = \prod_{j=1}^n b_{i_j} = \prod_{k=1}^K b_k^{c_k},$$

where $c_k$ is the count of the indices equals to $k$: $c_k = \sum_{j=1}^n \mathbf{1}(i_j = k)$. Now, we can substitute this into the expected value:

$$\mathbb{E}[b_{i_1} \ldots b_{i_n} f(\mathbf{b})] = \frac{1}{B(\boldsymbol{\alpha})} \int_{\mathcal{S}^K} f(\mathbf{b}) \left( \prod_{k=1}^K b_k^{c_k} \right) \left( \prod_{k=1}^K b_k^{\alpha_k - 1} \right) d\mathbf{b}$$

$$= \frac{1}{B(\boldsymbol{\alpha})} \int_{\mathcal{S}^K} f(\mathbf{b}) \prod_{k=1}^K b_k^{\alpha_k + c_k - 1} d\mathbf{b}$$

Define $\alpha'_k = \alpha_k + c_k$ and $\boldsymbol{\alpha}' = (\alpha'_1, \ldots, \alpha'_K)$. Let us rewrite the expression by multiplying and dividing by the normalization constant $B(\boldsymbol{\alpha}')$ for the $\text{Dir}(\boldsymbol{\alpha}')$ distribution:

$$\mathbb{E}[b_{i_1} \ldots b_{i_n} f(\mathbf{b})] = \frac{B(\boldsymbol{\alpha}')}{B(\boldsymbol{\alpha})} \int_{\mathcal{S}^K} f(\mathbf{b}) \frac{1}{B(\boldsymbol{\alpha}')} \prod_{k=1}^K b_k^{\alpha'_k - 1} d\mathbf{b}$$

$$= \frac{B(\boldsymbol{\alpha}')}{B(\boldsymbol{\alpha})} \int_{\mathcal{S}^K} f(\mathbf{b}) p(\mathbf{b}|\boldsymbol{\alpha}') d\mathbf{b},$$

where $p(\mathbf{b}|\boldsymbol{\alpha}')$ is the PDF of the $\text{Dir}(\boldsymbol{\alpha}')$ distribution. By substituting the known fact that $\frac{B(\boldsymbol{\alpha}')}{B(\boldsymbol{\alpha})} = \mathbb{E}[b_{i_1} \ldots b_{i_n}]$ (Balakrishnan & Nevzorov, 2004; Hoffmann, 2015), we get the desired result.

$\square$

Now, let us apply Lemma A.4 for the $i = j$ term:

$$\mathbb{E}[b_i^2(\log b_i)^2] = \mathbb{E}[b_i^2] \cdot \mathbb{E}''[(\log b_i'')^2]$$

where $\mathbf{b}'' \sim \mathrm{Dir}(\boldsymbol{\alpha} + 2\mathbf{e}_i)$. Now we apply Lemma A.2:

$$\mathbb{E}''[(\log b_i'')^2] = \big(\psi_1(\alpha_i + 2) - \psi_1(\alpha_0 + 2)\big) + \big(\psi(\alpha_i + 2) - \psi(\alpha_0 + 2)\big)^2$$

Since $\mathbb{E}[b_i^2] = \frac{\alpha_i(\alpha_i+1)}{\alpha_0(\alpha_0+1)}$, we get:

$$\mathbb{E}[b_i^2(\log b_i)^2] = \frac{\alpha_i(\alpha_i + 1)}{\alpha_0(\alpha_0 + 1)} \left\{ \big(\psi_1(\alpha_i + 2) - \psi_1(\alpha_0 + 2)\big) + \big(\psi(\alpha_i + 2) - \psi(\alpha_0 + 2)\big)^2 \right\}$$

Applying Lemma A.4 for the $i \neq j$ term:

$$\mathbb{E}[b_i b_j \log b_i \log b_j] = \mathbb{E}[b_i b_j] \cdot \mathbb{E}''[\log b_i'' \log b_j'']$$

where $\mathbf{b}'' \sim \mathrm{Dir}(\boldsymbol{\alpha} + \mathbf{e}_i + \mathbf{e}_j)$. Now, we apply Lemma A.3:

$$\mathbb{E}''[\log b_i'' \log b_j''] = -\psi_1(\alpha_0 + 2) + \big(\psi(\alpha_i + 1) - \psi(\alpha_0 + 2)\big)\big(\psi(\alpha_j + 1) - \psi(\alpha_0 + 2)\big)$$

Since $\mathbb{E}[b_i b_j] = \frac{\alpha_i \alpha_j}{\alpha_0(\alpha_0+1)}$ for $i \neq j$, we get:

$$\mathbb{E}[b_i b_j \log b_i \log b_j] = \frac{\alpha_i \alpha_j}{\alpha_0(\alpha_0 + 1)} \left\{ -\psi_1(\alpha_0 + 2) + \big(\psi(\alpha_i + 1) - \psi(\alpha_0 + 2)\big)\big(\psi(\alpha_j + 1) - \psi(\alpha_0 + 2)\big) \right\}$$

Combining these terms yields the final expression for $\mathbb{E}[\mathbf{h}^2]$:

$$\mathbb{E}[\mathbf{h}^2] = \frac{1}{\alpha_0(\alpha_0 + 1)} \left[ \sum_{i=1}^{K} \alpha_i(\alpha_i + 1) \left\{ \psi_1(\alpha_i + 2) - \psi_1(\alpha_0 + 2) + (\psi(\alpha_i + 2) - \psi(\alpha_0 + 2))^2 \right\} \right.$$

$$\left. + \sum_{i \neq j} \alpha_i \alpha_j \left\{ -\psi_1(\alpha_0 + 2) + (\psi(\alpha_i + 1) - \psi(\alpha_0 + 2))(\psi(\alpha_j + 1) - \psi(\alpha_0 + 2)) \right\} \right]$$

**Variance** $\mathrm{Var}[\mathbf{h}]$ The variance of the entropy under the Dirichlet distribution can then be computed using the standard formula:

$$\mathrm{Var}[\mathbf{h}] = \mathbb{E}[\mathbf{h}^2] - \mathbb{E}[\mathbf{h}]^2$$

using the analytical expressions for the first and second moments derived above.

### A.2 Mean and Variance of Expected Entropy under the Truncated Dirichlet Distribution

Recall from Section 3.2 that we define the truncated Dirichlet distribution

$$p_B^{\mathrm{trunc}}(\mathbf{b}; \mathcal{D}) = \begin{cases} \dfrac{p_B(\mathbf{b})}{Z(\mathcal{D})} & \text{if } \mathbf{b} \in \mathrm{constr}(\mathbf{b}, \mathcal{D}), \\ 0 & \text{otherwise}, \end{cases} \tag{9}$$

where $p_B(\mathbf{b})$ is the (untruncated) Dirichlet density with parameters $\boldsymbol{\alpha} + \mathbf{c}$, and

$$Z(\mathcal{D}) = \int_{\mathbf{b} \in \mathrm{constr}(\mathbf{b}, \mathcal{D})} p_B(\mathbf{b})\, d\mathbf{b}$$

is the (unknown) normalizing constant. Our goal is to compute the expected value and the variance of the entropy. To this end, we want to compute expectations of the form

$$\mathbb{E}\big[\mathbb{H}[\mathbf{b}]\big] \;=\; \int \mathbb{H}[\mathbf{b}]\, p_B^{\text{trunc}}(\mathbf{b};\mathcal{D})\,\mathrm{d}\mathbf{b}, \qquad \mathbb{E}\big[\mathbb{H}[\mathbf{b}]^2\big] \;=\; \int \mathbb{H}[\mathbf{b}]^2\, p_B^{\text{trunc}}(\mathbf{b};\mathcal{D})\,\mathrm{d}\mathbf{b}.$$

In the following, we focus only on the expected value of the entropy (the integral on the left), since the other integral will follow an analogous reasoning.

The first problem we face is that we cannot directly sample from $p_B^{\text{trunc}}(\mathbf{b};\mathcal{D})$. Therefore, we cannot approximate the integral via a simple Monte-Carlo approach. A naive solution would be standard *importance sampling* (IS) (Cochran, 1963; Owen, 2013). This approach consists of selecting a *proposal* distribution $q(\mathbf{b})$ with full support over the sample space, from which we know how to sample. Then, we apply some algebra to the original integral as follows:

$$\mathbb{E}[\mathbb{H}[\mathbf{b}]] \;=\; \int \mathbb{H}[\mathbf{b}]\, p_B^{\text{trunc}}(\mathbf{b};\mathcal{D})\,\mathrm{d}\mathbf{b} \;=\; \int \mathbb{H}[\mathbf{b}]\, \frac{p_B^{\text{trunc}}(\mathbf{b};\mathcal{D})}{q(\mathbf{b})} q(\mathbf{b})\,\mathrm{d}\mathbf{b} \;=\; \mathbb{E}_{\mathbf{b}\sim q}\left[ \frac{p_B^{\text{trunc}}(\mathbf{b};\mathcal{D})}{q(\mathbf{b})} \mathbb{H}[\mathbf{b}] \right].$$

As a proposal distribution, we could simply pick the uniform distribution over the truncated simplex, truncated according to $\text{constr}(\mathbf{b},\mathcal{D})$.

Now, we transformed the original expectation into an expectation with respect to a distribution from which we know how to sample. Therefore, we can apply Monte-Carlo to get an unbiased estimator. First, we sample $m$ samples from $q$. Then, we approximate the integral as follows:

$$\widehat{\mathbb{H}}_{\text{IS}} = \frac{1}{m}\sum_{i=1}^{m} \frac{p_B^{\text{trunc}}(\mathbf{b}_i;\mathcal{D})}{q(\mathbf{b}_i)} \mathbb{H}[\mathbf{b}_i]. \tag{10}$$

However, this estimator still requires knowledge of $Z(\mathcal{D})$ to compute $p_B^{\text{trunc}}(\mathbf{b}_i;\mathcal{D})$ for the importance weights, which we do not know in general. For this reason, in this paper we will use the *self-normalization* technique.

**Self-normalized importance sampling.** We can circumvent the explicit computation of $Z(\mathcal{D})$ by using *self-normalized* importance sampling (Owen, 2013; Swaminathan & Joachims, 2015). Self-normalized IS is similar to standard IS, but instead of dividing the sum by $m$, we use the sum of the importance weights:

$$\sum_{i=1}^{m} \frac{p_B^{\text{trunc}}(\mathbf{b}_i;\mathcal{D})}{q(\mathbf{b}_i)}\;.$$

Since the expected value of this quantity is $m$, this results in a biased-but-consistent estimator of the integral at hand (Swaminathan & Joachims, 2015), and in practice it has been found out to often provide a better estimation than simple IS.

After some algebraic simplifications, the integral approximation is computed as follows:

$$\widehat{\mathbb{H}}_{\text{SN}} = \frac{1}{\sum_{j=1}^{m} p_B(\mathbf{b}_j)} \sum_{i=1}^{m} p_B(\mathbf{b}_i)\mathbb{H}[\mathbf{b}_i]. \tag{11}$$

This simplified version follows from the following facts:

- we always sample from the truncated simplex, hence $p_B^{\text{trunc}}(\mathbf{b};\mathcal{D}) = \dfrac{p_B(\mathbf{b})}{Z(\mathcal{D})}$;

- the normalizing constant cancels out;

- the proposal distribution is constant and cancels out.

Now, this can be computed because we can sample from the truncated simplex and evaluate Dirichlet PDFs.

## B  Detailed Description of the Evaluation Methodology

We took the evaluation methodology of Farquhar et al. (2024) as a starting point and only modified it in ways which were necessary to adapt to our-use case. We describe the evaluation methodology in full in this section.

**Two-Stage Architecture**  The computation stage that does inference in the LLM (which takes over a week on a single A100 80GB) is separated from the stage that estimates semantic entropy (which only uses the CPU, taking on the order of 12 minutes). This is important so we don't have to repeat expensive GPU inference many times when changing details of the entropy estimation. Our code for the LLM inference stage is based on the code by Farquhar et al. (2024), while the code for the second stage is new.[11] We used the following quantization settings: 8 bit for Llama-3.3-70B, 16 bit for Mistral, 32 bit for Llama-3.2 and Llama-2.

**LLM Inference Stage**  Conservatively, we performed LLM inference for $N = 100$ times for each prompt. We use the temperature 1 for all generations. We also generate one extra answer with temperature 0.1, about which we aim to decide whether or not it is a hallucination. The source code for the LLM inference stage is based on the code by Farquhar et al. (2024). We found two bugs in the code, which we fixed. The first bug meant that the exact same LLM response (identical string) could be (rarely) assigned to different meaning classes, due to the imperfections of the DeBERTa entailment oracle. The second bug caused sum of probabilities of generated sequences to occasionally exceed one (it was caused by not storing the probabilities of special tokens ending the LLM response).

**Entropy Estimation Stage**  We implemented the entropy estimation stage as a single Jupyter notebook. In the second stage, when we require a dataset for a smaller value of $N$, we subsample (with replacement). The notebook can be used to regenerate all the tables and figures in the paper.

**Variable Budget**  Our biggest deviation from the methodology of Farquhar et al. (2024) is that we consider different choices of $N$ (the number of samples emitted by the LLM), where Farquhar et al. (2024) only considered the case of $N = 10$.

**Label Computation**  Even with supervised dataset, determining if a model hallucinates is not trivial because the LLM can phase the response in an arbitrary way. Following the work of Farquhar et al. (2024), the label determining if a model hallucinates, used for AUROC computation, was obtained by computing the F1 metric and thresholding it at 0.5.

**Confidence Bars**  All confidence bars for the AUROC estimates in our paper represent 1.96 times the standard error. Confidence bars are generated by resampling (with replacement) the dataset for a given value of $N$. In our tables, the bold font is applied as follows: the estimator with the best mean performance is put in bold, together with all the others with overlapping confidence bars.

## C  Additional Experimental Results

Below, we provide full experimental results (measured AUROC values) for a fixed budget choice (the number of samples from the LLM) of $N \in \{1, \dots, 10\}$.

---

[11]We will release the source code for both stages upon acceptance.

$N = 1$

| LLM | Dataset | LL | P(true) | SE-Bayes | SE-Histogram | SE-Rescaled | SE-Rescaled (h) |
|---|---|---|---|---|---|---|---|
| Llama-2 | NQ | $0.583 \pm 0.000$ | $0.461 \pm 0.000$ | $\mathbf{0.669 \pm 0.013}$ | $0.500 \pm 0.000$ | $0.500 \pm 0.000$ | $0.500 \pm 0.000$ |
| | SVAMP | $0.631 \pm 0.000$ | $0.469 \pm 0.000$ | $\mathbf{0.796 \pm 0.057}$ | $0.500 \pm 0.000$ | $0.500 \pm 0.000$ | $0.500 \pm 0.000$ |
| | Squad | $0.624 \pm 0.000$ | $0.441 \pm 0.000$ | $\mathbf{0.701 \pm 0.009}$ | $0.500 \pm 0.000$ | $0.500 \pm 0.000$ | $0.500 \pm 0.000$ |
| | Trivia QA | $0.594 \pm 0.000$ | $0.436 \pm 0.000$ | $\mathbf{0.688 \pm 0.012}$ | $0.500 \pm 0.000$ | $0.500 \pm 0.000$ | $0.500 \pm 0.000$ |
| Llama-3.2 | NQ | $0.627 \pm 0.000$ | $0.615 \pm 0.000$ | $\mathbf{0.683 \pm 0.012}$ | $0.500 \pm 0.000$ | $0.500 \pm 0.000$ | $0.500 \pm 0.000$ |
| | SVAMP | $0.647 \pm 0.000$ | $0.414 \pm 0.000$ | $\mathbf{0.812 \pm 0.012}$ | $0.500 \pm 0.000$ | $0.500 \pm 0.000$ | $0.500 \pm 0.000$ |
| | Squad | $\mathbf{0.610 \pm 0.000}$ | $0.555 \pm 0.000$ | $\mathbf{0.617 \pm 0.021}$ | $0.500 \pm 0.000$ | $0.500 \pm 0.000$ | $0.500 \pm 0.000$ |
| | Trivia QA | $0.614 \pm 0.000$ | $0.710 \pm 0.000$ | $\mathbf{0.731 \pm 0.008}$ | $0.500 \pm 0.000$ | $0.500 \pm 0.000$ | $0.500 \pm 0.000$ |
| Llama-3.3-70B | Trivia QA | $0.556 \pm 0.000$ | $0.686 \pm 0.000$ | $\mathbf{0.768 \pm 0.003}$ | $0.500 \pm 0.000$ | $0.500 \pm 0.000$ | $0.500 \pm 0.000$ |
| Mistral | NQ | $0.695 \pm 0.000$ | $\mathbf{0.731 \pm 0.000}$ | $0.635 \pm 0.008$ | $0.500 \pm 0.000$ | $0.500 \pm 0.000$ | $0.500 \pm 0.000$ |
| | SVAMP | $0.645 \pm 0.000$ | $\mathbf{0.843 \pm 0.000}$ | $0.844 \pm 0.038$ | $0.500 \pm 0.000$ | $0.500 \pm 0.000$ | $0.500 \pm 0.000$ |
| | Squad | $\mathbf{0.698 \pm 0.000}$ | $0.687 \pm 0.000$ | $0.628 \pm 0.010$ | $0.500 \pm 0.000$ | $0.500 \pm 0.000$ | $0.500 \pm 0.000$ |
| | Trivia QA | $\mathbf{0.672 \pm 0.000}$ | $0.647 \pm 0.000$ | $0.633 \pm 0.003$ | $0.500 \pm 0.000$ | $0.500 \pm 0.000$ | $0.500 \pm 0.000$ |

$N = 2$

| LLM | Dataset | LL | P(true) | SE-Bayes | SE-Histogram | SE-Rescaled | SE-Rescaled (h) |
|---|---|---|---|---|---|---|---|
| Llama-2 | NQ | $0.583 \pm 0.000$ | $0.461 \pm 0.000$ | $\mathbf{0.723 \pm 0.007}$ | $0.652 \pm 0.010$ | $0.654 \pm 0.013$ | $0.644 \pm 0.014$ |
| | SVAMP | $0.631 \pm 0.000$ | $0.469 \pm 0.000$ | $\mathbf{0.855 \pm 0.022}$ | $0.748 \pm 0.024$ | $0.749 \pm 0.027$ | $0.759 \pm 0.025$ |
| | Squad | $0.624 \pm 0.000$ | $0.441 \pm 0.000$ | $\mathbf{0.735 \pm 0.007}$ | $0.654 \pm 0.012$ | $0.659 \pm 0.017$ | $0.649 \pm 0.010$ |
| | Trivia QA | $0.594 \pm 0.000$ | $0.436 \pm 0.000$ | $\mathbf{0.737 \pm 0.006}$ | $0.670 \pm 0.012$ | $0.675 \pm 0.012$ | $0.673 \pm 0.010$ |
| Llama-3.2 | NQ | $0.627 \pm 0.000$ | $0.615 \pm 0.000$ | $\mathbf{0.728 \pm 0.008}$ | $0.640 \pm 0.013$ | $0.663 \pm 0.017$ | $0.649 \pm 0.020$ |
| | SVAMP | $0.647 \pm 0.000$ | $0.414 \pm 0.000$ | $\mathbf{0.845 \pm 0.022}$ | $0.761 \pm 0.032$ | $0.774 \pm 0.030$ | $0.771 \pm 0.028$ |
| | Squad | $0.610 \pm 0.000$ | $0.555 \pm 0.000$ | $\mathbf{0.664 \pm 0.019}$ | $0.608 \pm 0.024$ | $\mathbf{0.642 \pm 0.029}$ | $\mathbf{0.621 \pm 0.027}$ |
| | Trivia QA | $0.614 \pm 0.000$ | $0.710 \pm 0.000$ | $\mathbf{0.768 \pm 0.004}$ | $0.699 \pm 0.005$ | $0.706 \pm 0.007$ | $0.708 \pm 0.008$ |
| Llama-3.3-70B | Trivia QA | $0.556 \pm 0.000$ | $0.686 \pm 0.000$ | $\mathbf{0.775 \pm 0.003}$ | $0.653 \pm 0.006$ | $0.650 \pm 0.006$ | $0.651 \pm 0.006$ |
| Mistral | NQ | $0.695 \pm 0.000$ | $\mathbf{0.731 \pm 0.000}$ | $0.705 \pm 0.013$ | $0.647 \pm 0.007$ | $0.700 \pm 0.007$ | $0.654 \pm 0.009$ |
| | SVAMP | $0.645 \pm 0.000$ | $0.843 \pm 0.000$ | $\mathbf{0.876 \pm 0.011}$ | $0.793 \pm 0.023$ | $0.815 \pm 0.028$ | $0.812 \pm 0.024$ |
| | Squad | $\mathbf{0.698 \pm 0.000}$ | $0.687 \pm 0.000$ | $0.667 \pm 0.010$ | $0.618 \pm 0.005$ | $0.671 \pm 0.017$ | $0.631 \pm 0.010$ |
| | Trivia QA | $0.672 \pm 0.000$ | $0.647 \pm 0.000$ | $\mathbf{0.682 \pm 0.008}$ | $0.638 \pm 0.011$ | $0.645 \pm 0.013$ | $0.643 \pm 0.012$ |

$N = 3$

| LLM | Dataset | LL | P(true) | SE-Bayes | SE-Histogram | SE-Rescaled | SE-Rescaled (h) |
|---|---|---|---|---|---|---|---|
| Llama-2 | NQ | $0.583 \pm 0.000$ | $0.461 \pm 0.000$ | $\mathbf{0.746 \pm 0.006}$ | $0.707 \pm 0.009$ | $0.678 \pm 0.012$ | $0.700 \pm 0.011$ |
| | SVAMP | $0.631 \pm 0.000$ | $0.469 \pm 0.000$ | $\mathbf{0.862 \pm 0.020}$ | $0.810 \pm 0.028$ | $0.804 \pm 0.031$ | $0.811 \pm 0.031$ |
| | Squad | $0.624 \pm 0.000$ | $0.441 \pm 0.000$ | $\mathbf{0.754 \pm 0.013}$ | $0.710 \pm 0.015$ | $0.679 \pm 0.019$ | $0.708 \pm 0.012$ |
| | Trivia QA | $0.594 \pm 0.000$ | $0.436 \pm 0.000$ | $\mathbf{0.753 \pm 0.004}$ | $0.706 \pm 0.009$ | $0.710 \pm 0.007$ | $0.709 \pm 0.009$ |
| Llama-3.2 | NQ | $0.627 \pm 0.000$ | $0.615 \pm 0.000$ | $\mathbf{0.745 \pm 0.008}$ | $0.697 \pm 0.008$ | $0.672 \pm 0.012$ | $0.699 \pm 0.008$ |
| | SVAMP | $0.647 \pm 0.000$ | $0.414 \pm 0.000$ | $\mathbf{0.843 \pm 0.013}$ | $0.801 \pm 0.019$ | $0.800 \pm 0.021$ | $\mathbf{0.811 \pm 0.021}$ |
| | Squad | $0.610 \pm 0.000$ | $0.555 \pm 0.000$ | $\mathbf{0.686 \pm 0.021}$ | $\mathbf{0.659 \pm 0.016}$ | $\mathbf{0.649 \pm 0.017}$ | $\mathbf{0.654 \pm 0.014}$ |
| | Trivia QA | $0.614 \pm 0.000$ | $0.710 \pm 0.000$ | $\mathbf{0.781 \pm 0.008}$ | $0.738 \pm 0.006$ | $0.729 \pm 0.006$ | $0.746 \pm 0.007$ |
| Llama-3.3-70B | Trivia QA | $0.556 \pm 0.000$ | $0.686 \pm 0.000$ | $\mathbf{0.781 \pm 0.006}$ | $0.710 \pm 0.003$ | $0.702 \pm 0.004$ | $0.706 \pm 0.003$ |
| Mistral | NQ | $0.695 \pm 0.000$ | $0.731 \pm 0.000$ | $\mathbf{0.743 \pm 0.009}$ | $0.718 \pm 0.010$ | $0.716 \pm 0.009$ | $0.710 \pm 0.010$ |
| | SVAMP | $0.645 \pm 0.000$ | $0.843 \pm 0.000$ | $\mathbf{0.887 \pm 0.010}$ | $0.858 \pm 0.017$ | $0.851 \pm 0.021$ | $\mathbf{0.870 \pm 0.022}$ |
| | Squad | $\mathbf{0.698 \pm 0.000}$ | $0.687 \pm 0.000$ | $\mathbf{0.694 \pm 0.010}$ | $0.676 \pm 0.018$ | $\mathbf{0.698 \pm 0.014}$ | $\mathbf{0.685 \pm 0.020}$ |
| | Trivia QA | $0.672 \pm 0.000$ | $0.647 \pm 0.000$ | $\mathbf{0.691 \pm 0.007}$ | $\mathbf{0.678 \pm 0.010}$ | $0.676 \pm 0.008$ | $\mathbf{0.684 \pm 0.010}$ |

$N = 4$

| LLM | Dataset | LL | P(true) | SE-Bayes | SE-Histogram | SE-Rescaled | SE-Rescaled (h) |
|---|---|---|---|---|---|---|---|
| Llama-2 | NQ | $0.583 \pm 0.000$ | $0.461 \pm 0.000$ | $\mathbf{0.754 \pm 0.002}$ | $0.729 \pm 0.007$ | $0.695 \pm 0.014$ | $0.720 \pm 0.008$ |
| | SVAMP | $0.631 \pm 0.000$ | $0.469 \pm 0.000$ | $\mathbf{0.899 \pm 0.013}$ | $0.866 \pm 0.010$ | $0.870 \pm 0.015$ | $0.872 \pm 0.011$ |
| | Squad | $0.624 \pm 0.000$ | $0.441 \pm 0.000$ | $\mathbf{0.771 \pm 0.009}$ | $0.742 \pm 0.006$ | $0.700 \pm 0.008$ | $0.734 \pm 0.004$ |
| | Trivia QA | $0.594 \pm 0.000$ | $0.436 \pm 0.000$ | $\mathbf{0.767 \pm 0.003}$ | $0.735 \pm 0.007$ | $0.733 \pm 0.007$ | $0.734 \pm 0.007$ |
| Llama-3.2 | NQ | $0.627 \pm 0.000$ | $0.615 \pm 0.000$ | $\mathbf{0.759 \pm 0.007}$ | $0.722 \pm 0.006$ | $0.684 \pm 0.015$ | $0.723 \pm 0.007$ |
| | SVAMP | $0.647 \pm 0.000$ | $0.414 \pm 0.000$ | $\mathbf{0.868 \pm 0.005}$ | $0.843 \pm 0.005$ | $0.837 \pm 0.010$ | $0.850 \pm 0.003$ |
| | Squad | $0.610 \pm 0.000$ | $0.555 \pm 0.000$ | $\mathbf{0.686 \pm 0.015}$ | $\mathbf{0.686 \pm 0.014}$ | $0.649 \pm 0.019$ | $\mathbf{0.676 \pm 0.018}$ |
| | Trivia QA | $0.614 \pm 0.000$ | $0.710 \pm 0.000$ | $\mathbf{0.786 \pm 0.007}$ | $0.759 \pm 0.009$ | $0.746 \pm 0.005$ | $0.764 \pm 0.009$ |
| Llama-3.3-70B | Trivia QA | $0.556 \pm 0.000$ | $0.686 \pm 0.000$ | $\mathbf{0.793 \pm 0.002}$ | $0.739 \pm 0.002$ | $0.734 \pm 0.003$ | $0.738 \pm 0.002$ |
| Mistral | NQ | $0.695 \pm 0.000$ | $0.731 \pm 0.000$ | $\mathbf{0.762 \pm 0.012}$ | $0.740 \pm 0.008$ | $0.730 \pm 0.006$ | $0.737 \pm 0.013$ |
| | SVAMP | $0.645 \pm 0.000$ | $0.843 \pm 0.000$ | $\mathbf{0.880 \pm 0.015}$ | $0.865 \pm 0.017$ | $0.851 \pm 0.019$ | $\mathbf{0.877 \pm 0.019}$ |
| | Squad | $0.698 \pm 0.000$ | $0.687 \pm 0.000$ | $\mathbf{0.720 \pm 0.007}$ | $\mathbf{0.705 \pm 0.011}$ | $0.700 \pm 0.014$ | $\mathbf{0.705 \pm 0.013}$ |
| | Trivia QA | $0.672 \pm 0.000$ | $0.647 \pm 0.000$ | $\mathbf{0.692 \pm 0.009}$ | $\mathbf{0.682 \pm 0.016}$ | $\mathbf{0.678 \pm 0.013}$ | $\mathbf{0.687 \pm 0.012}$ |

$N = 5$

| LLM | Dataset | LL | P(true) | SE-Bayes | SE-Histogram | SE-Rescaled | SE-Rescaled (h) |
|---|---|---|---|---|---|---|---|
| Llama-2 | NQ | $0.583 \pm 0.000$ | $0.461 \pm 0.000$ | $\mathbf{0.752 \pm 0.006}$ | $0.730 \pm 0.009$ | $0.701 \pm 0.012$ | $0.725 \pm 0.010$ |
| | SVAMP | $0.631 \pm 0.000$ | $0.469 \pm 0.000$ | $\mathbf{0.871 \pm 0.011}$ | $\mathbf{0.849 \pm 0.012}$ | $\mathbf{0.853 \pm 0.013}$ | $\mathbf{0.858 \pm 0.013}$ |
| | Squad | $0.624 \pm 0.000$ | $0.441 \pm 0.000$ | $\mathbf{0.774 \pm 0.008}$ | $0.756 \pm 0.004$ | $0.711 \pm 0.012$ | $0.752 \pm 0.005$ |
| | Trivia QA | $0.594 \pm 0.000$ | $0.436 \pm 0.000$ | $\mathbf{0.763 \pm 0.009}$ | $0.734 \pm 0.007$ | $0.737 \pm 0.006$ | $0.735 \pm 0.006$ |
| Llama-3.2 | NQ | $0.627 \pm 0.000$ | $0.615 \pm 0.000$ | $\mathbf{0.760 \pm 0.008}$ | $0.732 \pm 0.012$ | $0.691 \pm 0.005$ | $0.733 \pm 0.012$ |
| | SVAMP | $0.647 \pm 0.000$ | $0.414 \pm 0.000$ | $\mathbf{0.870 \pm 0.009}$ | $\mathbf{0.860 \pm 0.010}$ | $0.850 \pm 0.004$ | $\mathbf{0.864 \pm 0.009}$ |
| | Squad | $0.610 \pm 0.000$ | $0.555 \pm 0.000$ | $\mathbf{0.710 \pm 0.017}$ | $\mathbf{0.707 \pm 0.012}$ | $0.667 \pm 0.021$ | $\mathbf{0.705 \pm 0.013}$ |
| | Trivia QA | $0.614 \pm 0.000$ | $0.710 \pm 0.000$ | $\mathbf{0.792 \pm 0.004}$ | $0.775 \pm 0.002$ | $0.763 \pm 0.005$ | $0.777 \pm 0.004$ |
| Llama-3.3-70B | Trivia QA | $0.556 \pm 0.000$ | $0.686 \pm 0.000$ | $\mathbf{0.793 \pm 0.005}$ | $0.750 \pm 0.005$ | $0.740 \pm 0.004$ | $0.747 \pm 0.006$ |
| Mistral | NQ | $0.695 \pm 0.000$ | $0.731 \pm 0.000$ | $\mathbf{0.780 \pm 0.006}$ | $0.762 \pm 0.007$ | $0.728 \pm 0.008$ | $0.756 \pm 0.007$ |
| | SVAMP | $0.645 \pm 0.000$ | $0.843 \pm 0.000$ | $\mathbf{0.880 \pm 0.019}$ | $0.866 \pm 0.021$ | $0.855 \pm 0.027$ | $\mathbf{0.878 \pm 0.022}$ |
| | Squad | $0.698 \pm 0.000$ | $0.687 \pm 0.000$ | $\mathbf{0.731 \pm 0.008}$ | $\mathbf{0.719 \pm 0.010}$ | $0.699 \pm 0.008$ | $0.712 \pm 0.008$ |
| | Trivia QA | $0.672 \pm 0.000$ | $0.647 \pm 0.000$ | $\mathbf{0.691 \pm 0.005}$ | $\mathbf{0.688 \pm 0.005}$ | $\mathbf{0.684 \pm 0.004}$ | $\mathbf{0.690 \pm 0.005}$ |

$N = 6$

| LLM | Dataset | LL | P(true) | SE-Bayes | SE-Histogram | SE-Rescaled | SE-Rescaled (h) |
|---|---|---|---|---|---|---|---|
| Llama-2 | NQ | $0.583 \pm 0.000$ | $0.461 \pm 0.000$ | $\mathbf{0.755 \pm 0.005}$ | $0.739 \pm 0.010$ | $0.702 \pm 0.008$ | $0.733 \pm 0.010$ |
| | SVAMP | $0.631 \pm 0.000$ | $0.469 \pm 0.000$ | $\mathbf{0.892 \pm 0.013}$ | $\mathbf{0.876 \pm 0.018}$ | $\mathbf{0.876 \pm 0.017}$ | $\mathbf{0.880 \pm 0.016}$ |
| | Squad | $0.624 \pm 0.000$ | $0.441 \pm 0.000$ | $\mathbf{0.776 \pm 0.005}$ | $0.761 \pm 0.009$ | $0.725 \pm 0.006$ | $0.756 \pm 0.009$ |
| | Trivia QA | $0.594 \pm 0.000$ | $0.436 \pm 0.000$ | $\mathbf{0.771 \pm 0.006}$ | $0.743 \pm 0.009$ | $0.747 \pm 0.008$ | $0.745 \pm 0.009$ |
| Llama-3.2 | NQ | $0.627 \pm 0.000$ | $0.615 \pm 0.000$ | $\mathbf{0.765 \pm 0.005}$ | $0.741 \pm 0.004$ | $0.690 \pm 0.012$ | $0.738 \pm 0.006$ |
| | SVAMP | $0.647 \pm 0.000$ | $0.414 \pm 0.000$ | $\mathbf{0.862 \pm 0.007}$ | $0.844 \pm 0.009$ | $0.833 \pm 0.010$ | $0.844 \pm 0.011$ |
| | Squad | $0.610 \pm 0.000$ | $0.555 \pm 0.000$ | $\mathbf{0.716 \pm 0.006}$ | $\mathbf{0.718 \pm 0.010}$ | $0.671 \pm 0.012$ | $\mathbf{0.715 \pm 0.009}$ |
| | Trivia QA | $0.614 \pm 0.000$ | $0.710 \pm 0.000$ | $\mathbf{0.795 \pm 0.005}$ | $\mathbf{0.783 \pm 0.008}$ | $0.771 \pm 0.008$ | $\mathbf{0.784 \pm 0.008}$ |
| Llama-3.3-70B | Trivia QA | $0.556 \pm 0.000$ | $0.686 \pm 0.000$ | $\mathbf{0.798 \pm 0.003}$ | $0.759 \pm 0.006$ | $0.751 \pm 0.004$ | $0.756 \pm 0.007$ |
| Mistral | NQ | $0.695 \pm 0.000$ | $0.731 \pm 0.000$ | $\mathbf{0.774 \pm 0.010}$ | $\mathbf{0.763 \pm 0.009}$ | $0.740 \pm 0.008$ | $\mathbf{0.756 \pm 0.008}$ |
| | SVAMP | $0.645 \pm 0.000$ | $0.843 \pm 0.000$ | $\mathbf{0.888 \pm 0.020}$ | $0.878 \pm 0.023$ | $0.860 \pm 0.018$ | $\mathbf{0.885 \pm 0.018}$ |
| | Squad | $0.698 \pm 0.000$ | $0.687 \pm 0.000$ | $\mathbf{0.726 \pm 0.004}$ | $\mathbf{0.721 \pm 0.006}$ | $0.700 \pm 0.011$ | $0.719 \pm 0.002$ |
| | Trivia QA | $0.672 \pm 0.000$ | $0.647 \pm 0.000$ | $\mathbf{0.691 \pm 0.010}$ | $\mathbf{0.698 \pm 0.012}$ | $\mathbf{0.690 \pm 0.011}$ | $\mathbf{0.697 \pm 0.011}$ |

$N = 7$

| LLM | Dataset | LL | P(true) | SE-Bayes | SE-Histogram | SE-Rescaled | SE-Rescaled (h) |
|---|---|---|---|---|---|---|---|
| Llama-2 | NQ | $0.583 \pm 0.000$ | $0.461 \pm 0.000$ | $\mathbf{0.760 \pm 0.002}$ | $0.744 \pm 0.004$ | $0.712 \pm 0.004$ | $0.738 \pm 0.004$ |
| | SVAMP | $0.631 \pm 0.000$ | $0.469 \pm 0.000$ | $\mathbf{0.895 \pm 0.016}$ | $\mathbf{0.875 \pm 0.018}$ | $\mathbf{0.875 \pm 0.017}$ | $\mathbf{0.880 \pm 0.017}$ |
| | Squad | $0.624 \pm 0.000$ | $0.441 \pm 0.000$ | $\mathbf{0.778 \pm 0.004}$ | $0.767 \pm 0.004$ | $0.718 \pm 0.007$ | $0.763 \pm 0.005$ |
| | Trivia QA | $0.594 \pm 0.000$ | $0.436 \pm 0.000$ | $\mathbf{0.767 \pm 0.003}$ | $0.750 \pm 0.006$ | $0.752 \pm 0.006$ | $0.751 \pm 0.006$ |
| Llama-3.2 | NQ | $0.627 \pm 0.000$ | $0.615 \pm 0.000$ | $\mathbf{0.770 \pm 0.006}$ | $0.750 \pm 0.007$ | $0.697 \pm 0.009$ | $0.747 \pm 0.006$ |
| | SVAMP | $0.647 \pm 0.000$ | $0.414 \pm 0.000$ | $\mathbf{0.862 \pm 0.011}$ | $\mathbf{0.847 \pm 0.019}$ | $\mathbf{0.858 \pm 0.020}$ | $\mathbf{0.847 \pm 0.020}$ |
| | Squad | $0.610 \pm 0.000$ | $0.555 \pm 0.000$ | $\mathbf{0.723 \pm 0.006}$ | $\mathbf{0.725 \pm 0.006}$ | $0.667 \pm 0.008$ | $\mathbf{0.719 \pm 0.006}$ |
| | Trivia QA | $0.614 \pm 0.000$ | $0.710 \pm 0.000$ | $\mathbf{0.795 \pm 0.001}$ | $0.782 \pm 0.004$ | $0.774 \pm 0.008$ | $0.783 \pm 0.003$ |
| Llama-3.3-70B | Trivia QA | $0.556 \pm 0.000$ | $0.686 \pm 0.000$ | $\mathbf{0.796 \pm 0.004}$ | $0.765 \pm 0.006$ | $0.756 \pm 0.005$ | $0.763 \pm 0.007$ |
| Mistral | NQ | $0.695 \pm 0.000$ | $0.731 \pm 0.000$ | $\mathbf{0.782 \pm 0.003}$ | $0.770 \pm 0.005$ | $0.736 \pm 0.005$ | $0.764 \pm 0.005$ |
| | SVAMP | $0.645 \pm 0.000$ | $0.843 \pm 0.000$ | $\mathbf{0.907 \pm 0.005}$ | $0.903 \pm 0.001$ | $0.890 \pm 0.015$ | $\mathbf{0.910 \pm 0.004}$ |
| | Squad | $0.698 \pm 0.000$ | $0.687 \pm 0.000$ | $\mathbf{0.739 \pm 0.008}$ | $\mathbf{0.735 \pm 0.008}$ | $0.702 \pm 0.010$ | $\mathbf{0.728 \pm 0.007}$ |
| | Trivia QA | $0.672 \pm 0.000$ | $0.647 \pm 0.000$ | $\mathbf{0.697 \pm 0.011}$ | $\mathbf{0.702 \pm 0.007}$ | $\mathbf{0.696 \pm 0.009}$ | $\mathbf{0.702 \pm 0.009}$ |

$N = 8$

| LLM | Dataset | LL | P(true) | SE-Bayes | SE-Histogram | SE-Rescaled | SE-Rescaled (h) |
|---|---|---|---|---|---|---|---|
| Llama-2 | NQ | $0.583 \pm 0.000$ | $0.461 \pm 0.000$ | $\mathbf{0.761 \pm 0.005}$ | $0.749 \pm 0.006$ | $0.722 \pm 0.010$ | $0.742 \pm 0.007$ |
| | SVAMP | $0.631 \pm 0.000$ | $0.469 \pm 0.000$ | $\mathbf{0.900 \pm 0.002}$ | $0.888 \pm 0.007$ | $0.882 \pm 0.010$ | $0.889 \pm 0.007$ |
| | Squad | $0.624 \pm 0.000$ | $0.441 \pm 0.000$ | $\mathbf{0.784 \pm 0.007}$ | $\mathbf{0.778 \pm 0.006}$ | $0.734 \pm 0.015$ | $\mathbf{0.775 \pm 0.008}$ |
| | Trivia QA | $0.594 \pm 0.000$ | $0.436 \pm 0.000$ | $\mathbf{0.778 \pm 0.002}$ | $0.761 \pm 0.007$ | $0.762 \pm 0.007$ | $0.760 \pm 0.007$ |
| Llama-3.2 | NQ | $0.627 \pm 0.000$ | $0.615 \pm 0.000$ | $\mathbf{0.773 \pm 0.005}$ | $0.753 \pm 0.005$ | $0.704 \pm 0.004$ | $0.748 \pm 0.006$ |
| | SVAMP | $0.647 \pm 0.000$ | $0.414 \pm 0.000$ | $\mathbf{0.871 \pm 0.014}$ | $\mathbf{0.868 \pm 0.013}$ | $\mathbf{0.862 \pm 0.012}$ | $\mathbf{0.868 \pm 0.011}$ |
| | Squad | $0.610 \pm 0.000$ | $0.555 \pm 0.000$ | $\mathbf{0.724 \pm 0.009}$ | $\mathbf{0.724 \pm 0.008}$ | $0.668 \pm 0.014$ | $\mathbf{0.720 \pm 0.009}$ |
| | Trivia QA | $0.614 \pm 0.000$ | $0.710 \pm 0.000$ | $\mathbf{0.797 \pm 0.001}$ | $0.785 \pm 0.002$ | $0.778 \pm 0.004$ | $0.787 \pm 0.002$ |
| Llama-3.3-70B | Trivia QA | $0.556 \pm 0.000$ | $0.686 \pm 0.000$ | $\mathbf{0.798 \pm 0.005}$ | $0.772 \pm 0.004$ | $0.764 \pm 0.005$ | $0.770 \pm 0.003$ |
| Mistral | NQ | $0.695 \pm 0.000$ | $0.731 \pm 0.000$ | $\mathbf{0.791 \pm 0.004}$ | $\mathbf{0.784 \pm 0.005}$ | $0.742 \pm 0.005$ | $0.775 \pm 0.005$ |
| | SVAMP | $0.645 \pm 0.000$ | $0.843 \pm 0.000$ | $\mathbf{0.892 \pm 0.014}$ | $\mathbf{0.888 \pm 0.015}$ | $\mathbf{0.882 \pm 0.012}$ | $\mathbf{0.893 \pm 0.016}$ |
| | Squad | $0.698 \pm 0.000$ | $0.687 \pm 0.000$ | $\mathbf{0.744 \pm 0.005}$ | $\mathbf{0.745 \pm 0.005}$ | $0.710 \pm 0.004$ | $\mathbf{0.737 \pm 0.005}$ |
| | Trivia QA | $0.672 \pm 0.000$ | $0.647 \pm 0.000$ | $\mathbf{0.699 \pm 0.003}$ | $\mathbf{0.702 \pm 0.006}$ | $\mathbf{0.701 \pm 0.010}$ | $\mathbf{0.704 \pm 0.008}$ |

$N = 9$

| LLM | Dataset | LL | P(true) | SE-Bayes | SE-Histogram | SE-Rescaled | SE-Rescaled (h) |
|---|---|---|---|---|---|---|---|
| Llama-2 | NQ | $0.583 \pm 0.000$ | $0.461 \pm 0.000$ | $\mathbf{0.759 \pm 0.005}$ | $\mathbf{0.749 \pm 0.008}$ | $0.723 \pm 0.008$ | $0.744 \pm 0.009$ |
| | SVAMP | $0.631 \pm 0.000$ | $0.469 \pm 0.000$ | $\mathbf{0.894 \pm 0.005}$ | $0.877 \pm 0.007$ | $0.879 \pm 0.006$ | $0.880 \pm 0.007$ |
| | Squad | $0.624 \pm 0.000$ | $0.441 \pm 0.000$ | $\mathbf{0.790 \pm 0.008}$ | $\mathbf{0.785 \pm 0.010}$ | $0.738 \pm 0.010$ | $\mathbf{0.780 \pm 0.011}$ |
| | Trivia QA | $0.594 \pm 0.000$ | $0.436 \pm 0.000$ | $\mathbf{0.780 \pm 0.004}$ | $0.756 \pm 0.008$ | $0.758 \pm 0.009$ | $0.755 \pm 0.006$ |
| Llama-3.2 | NQ | $0.627 \pm 0.000$ | $0.615 \pm 0.000$ | $\mathbf{0.770 \pm 0.004}$ | $0.753 \pm 0.006$ | $0.708 \pm 0.007$ | $0.747 \pm 0.005$ |
| | SVAMP | $0.647 \pm 0.000$ | $0.414 \pm 0.000$ | $\mathbf{0.875 \pm 0.006}$ | $\mathbf{0.876 \pm 0.010}$ | $\mathbf{0.880 \pm 0.012}$ | $\mathbf{0.875 \pm 0.010}$ |
| | Squad | $0.610 \pm 0.000$ | $0.555 \pm 0.000$ | $\mathbf{0.732 \pm 0.010}$ | $\mathbf{0.733 \pm 0.011}$ | $0.674 \pm 0.010$ | $\mathbf{0.727 \pm 0.012}$ |
| | Trivia QA | $0.614 \pm 0.000$ | $0.710 \pm 0.000$ | $\mathbf{0.800 \pm 0.004}$ | $\mathbf{0.793 \pm 0.006}$ | $0.788 \pm 0.007$ | $\mathbf{0.794 \pm 0.005}$ |
| Llama-3.3-70B | Trivia QA | $0.556 \pm 0.000$ | $0.686 \pm 0.000$ | $\mathbf{0.794 \pm 0.002}$ | $0.770 \pm 0.006$ | $0.762 \pm 0.006$ | $0.769 \pm 0.005$ |
| Mistral | NQ | $0.695 \pm 0.000$ | $0.731 \pm 0.000$ | $\mathbf{0.795 \pm 0.005}$ | $0.785 \pm 0.004$ | $0.752 \pm 0.008$ | $0.777 \pm 0.004$ |
| | SVAMP | $0.645 \pm 0.000$ | $0.843 \pm 0.000$ | $\mathbf{0.903 \pm 0.007}$ | $\mathbf{0.895 \pm 0.010}$ | $0.877 \pm 0.015$ | $\mathbf{0.896 \pm 0.009}$ |
| | Squad | $0.698 \pm 0.000$ | $0.687 \pm 0.000$ | $\mathbf{0.749 \pm 0.006}$ | $\mathbf{0.747 \pm 0.007}$ | $0.717 \pm 0.008$ | $\mathbf{0.739 \pm 0.008}$ |
| | Trivia QA | $0.672 \pm 0.000$ | $0.647 \pm 0.000$ | $\mathbf{0.698 \pm 0.009}$ | $\mathbf{0.704 \pm 0.012}$ | $\mathbf{0.702 \pm 0.013}$ | $\mathbf{0.704 \pm 0.013}$ |

| | | | | $N = 10$ | | | |
|---|---|---|---|---|---|---|---|
| LLM | Dataset | LL | P(true) | SE-Bayes | SE-Histogram | SE-Rescaled | SE-Rescaled (h) |
| Llama-2 | NQ | $0.583 \pm 0.000$ | $0.461 \pm 0.000$ | $\mathbf{0.763 \pm 0.003}$ | $0.753 \pm 0.002$ | $0.731 \pm 0.007$ | $0.745 \pm 0.002$ |
| | SVAMP | $0.631 \pm 0.000$ | $0.469 \pm 0.000$ | $\mathbf{0.896 \pm 0.007}$ | $\mathbf{0.886 \pm 0.006}$ | $\mathbf{0.891 \pm 0.007}$ | $\mathbf{0.890 \pm 0.010}$ |
| | Squad | $0.624 \pm 0.000$ | $0.441 \pm 0.000$ | $\mathbf{0.785 \pm 0.003}$ | $\mathbf{0.782 \pm 0.006}$ | $0.735 \pm 0.005$ | $0.776 \pm 0.005$ |
| | Trivia QA | $0.594 \pm 0.000$ | $0.436 \pm 0.000$ | $\mathbf{0.776 \pm 0.003}$ | $0.761 \pm 0.004$ | $0.765 \pm 0.004$ | $0.760 \pm 0.004$ |
| Llama-3.2 | NQ | $0.627 \pm 0.000$ | $0.615 \pm 0.000$ | $\mathbf{0.777 \pm 0.004}$ | $0.761 \pm 0.004$ | $0.711 \pm 0.007$ | $0.756 \pm 0.005$ |
| | SVAMP | $0.647 \pm 0.000$ | $0.414 \pm 0.000$ | $\mathbf{0.863 \pm 0.011}$ | $\mathbf{0.861 \pm 0.010}$ | $\mathbf{0.856 \pm 0.010}$ | $\mathbf{0.859 \pm 0.012}$ |
| | Squad | $0.610 \pm 0.000$ | $0.555 \pm 0.000$ | $\mathbf{0.737 \pm 0.007}$ | $\mathbf{0.736 \pm 0.008}$ | $0.686 \pm 0.008$ | $\mathbf{0.729 \pm 0.008}$ |
| | Trivia QA | $0.614 \pm 0.000$ | $0.710 \pm 0.000$ | $\mathbf{0.801 \pm 0.001}$ | $0.793 \pm 0.002$ | $0.791 \pm 0.004$ | $0.793 \pm 0.002$ |
| Llama-3.3-70B | Trivia QA | $0.556 \pm 0.000$ | $0.686 \pm 0.000$ | $\mathbf{0.798 \pm 0.004}$ | $0.777 \pm 0.002$ | $0.771 \pm 0.003$ | $0.777 \pm 0.003$ |
| Mistral | NQ | $0.695 \pm 0.000$ | $0.731 \pm 0.000$ | $\mathbf{0.796 \pm 0.004}$ | $\mathbf{0.790 \pm 0.004}$ | $0.758 \pm 0.006$ | $0.779 \pm 0.004$ |
| | SVAMP | $0.645 \pm 0.000$ | $0.843 \pm 0.000$ | $\mathbf{0.915 \pm 0.012}$ | $\mathbf{0.916 \pm 0.011}$ | $0.885 \pm 0.015$ | $\mathbf{0.919 \pm 0.010}$ |
| | Squad | $0.698 \pm 0.000$ | $0.687 \pm 0.000$ | $\mathbf{0.752 \pm 0.006}$ | $\mathbf{0.749 \pm 0.007}$ | $0.714 \pm 0.002$ | $\mathbf{0.741 \pm 0.006}$ |
| | Trivia QA | $0.672 \pm 0.000$ | $0.647 \pm 0.000$ | $\mathbf{0.697 \pm 0.002}$ | $\mathbf{0.702 \pm 0.009}$ | $0.697 \pm 0.008$ | $\mathbf{0.702 \pm 0.009}$ |

## D  Hyperparameter Sensitivity

We examined the sensitivity of our method to the hyperparameter $\alpha$, finding it to be insensitive. The plots below show performance for Llama 2 on the Trivia QA dataset for $\alpha \in \{0.1, 0.5, 1.0\}$. A value of 0.5 was used for all other experiments.

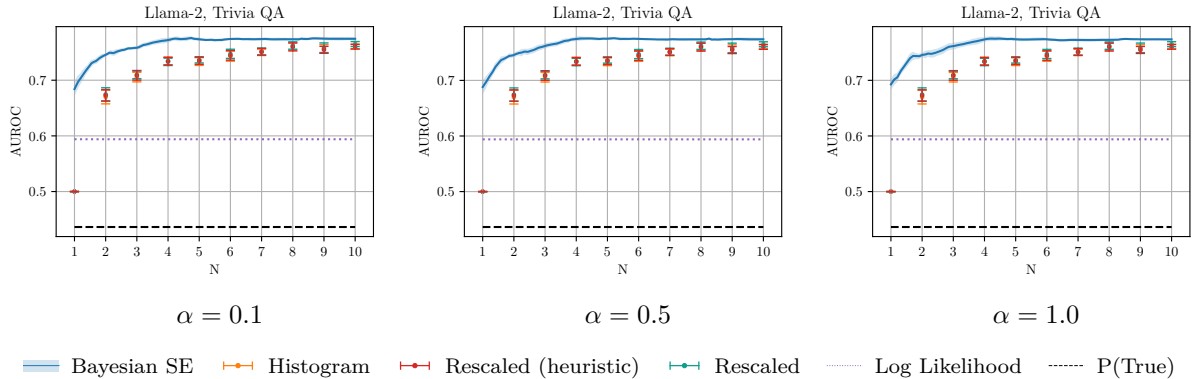

