# OpenReview forum: "Hallucination Detection on a Budget: Efficient Bayesian Estimation of Semantic Entropy"
_TMLR — Accepted by TMLR_

### Review · Reviewer_iPCg · 2025-05-18

**Summary Of Contributions:**

This paper suggests an improved version of the semantic entropy estimation for LLM outputs. Based on an earlier study, the authors claim that hallucinations can be detected based on the entropy of the generated responses in the semantic domain by mapping responses into meaning. This paper replaces the large scale sampling and histogram based approach of the referred paper by a Bayesian estimation approach. The experiments show the estimated entropy values and the area under ROC curve for hallucination detection on a few question answering tasks using Llama3-2b model. Based on the figures in Fig. 1 and and Fig. 2 especially in the case where the estimation process uses small number of generations (N < 4), the Bayesian estimation technique proposed in the paper provides better estimates. However, when N is large the difference between different entropy estimation methods gets smaller.

**Audience:**

Yes

**Broader Impact Concerns:**

There is not an explicit section on ethical implications. The paper tries to detect LLM hallucinations and hence can be useful for eliminating some of the potential harms of LLMs.

**Claims And Evidence:**

Yes

**Requested Changes:**

1. The paper refers to (Farquhar et al., 2024) for some critical details such as the relationship between LLM hallucinations and the entropy. It could be useful to explain it at least in a few sentences.

2. The experimental setup can also be explained in more details

3. In Table 1, do the numbers represent the expected value of the entropy? If so, instead of just writing "Experimental results " or saying performance, please clarify the Table 1 caption and explain why the higher mean entropy means a better estimator?

4. Some analysis on the length of the LLM responses and whether and how they affect the entropy estimation can be discussed.

**Strengths And Weaknesses:**

Strengths:
- Relatively clearly written. Formulations seem correct.


Weaknesses:
- The results are not surprising as Bayesian estimation tends to outperform maximum likelihood style solutions in cases where there is small number of samples. Hence, it is not clear what the novelty of this paper is other than applying the above idea to the particular problem of LLM hallucination detection.
- The paper is also not very self contained in the sense that the paper refers to (Farquhar et al., 2024) for some critical details such as the relationship between LLM hallucinations and the entropy. It could be useful to explain it at least in a few sentences.

---

> ### Author Response · Authors · 2025-05-19
> **Author Response**
>
> Thank you for the review. First, let us comment on the identified weaknesses.
>
> 1. We will update the paper explaining the relationship between LLM hallucinations and the semantic entropy more. The general idea is that higher entropy means that the model is more likely to hallucinate.
>
> 2. The experimental setup is currently explained in the supplementary material in a way where we emphasise the difference between what we do and the paper by Farquhar et al. (2024). We will provide a more self-contained description of the setup in the revised version. In addition, we will release the source code upon acceptance.
>
> 3. The numbers in Table 1 represent AUROC (area under the ROC curve). Higher numbers imply better performance in identifying hallucinations. We will make this clear in a revised version of the paper.
>
> 4. Examining how the length of LLM responses affects entropy estimation currently looks to us out of scope for the paper. Our goal is to improve sample efficiency i.e. to do what Farquhar et al. (2024) has done, only with way fewer LLM samples. Correlating the length of LLM responses and the quality of hallucination detection is an interesting but orthogonal research question. An experiment addressing this question would likely need to be done with a large budget, to separate two concerns: sampling noise and the question of how well the entailment oracle setup still works for very long answers. If you are unconvinced by this argument, we can include results computing AUROC vs a given response length in a revised version. *Please do let us know if you require this.*
>
> We also wanted to make a couple of general comments.
> - You mention "when N is large the difference between different entropy estimation methods gets smaller”. This is true - in fact any correctly implemented consistent estimator of semantic entropy will converge to the same value for a large sample budget, implying diminishing difference between estimators for large N. However, for small N, differences between estimators will be huge. In our paper, we emphasise the setting of small N because of its huge practical relevance - LLM inference is still very expensive.
> - You say the novelty of the paper is limited and the result is not surprising. We believe that achieving SOTA sample efficiency on the problem of measuring semantic entropy is a worthy contribution, of interest to the  TMLR community. Also, we stress that the mechanics of how we condition on the probability of the generated sequence being lower bounded by the observed probability is entirely novel - we are not aware of any other entropy estimation approach doing this.

---

### Review · Reviewer_jKjg · 2025-05-22

**Summary Of Contributions:**

This paper introduces an method for approximating semantic entropy using posterior inference on a meaning distribution. This distribution implies a distribution on the entropy of the prompts and can be approximated well using substantially less generated data from an LLM. The results compare favourably with baselines from the original paper as well as other approaches used for detecting hallucinations in LLMs.

**Audience:**

Yes

**Broader Impact Concerns:**

I have no ethical concerns about this work.

**Claims And Evidence:**

Yes

**Requested Changes:**

It would be nice to explain why the approximation more accurately predicts a hallucination than the more expensive estimate of semantic entropy. Why do they not converge as N is made larger?

**Strengths And Weaknesses:**

The paper is well written and makes a meaningful and significant contribution. The experiments are well thought out and convincingly show that the method works. It would make this paper stronger if there could be some explanation for why it works better at least against some of the baselines. There is little explanation given for even an intuition for why these results should hold and a supporting experiment.

---

> ### Author Response · Authors · 2025-05-22
> **Autor Response**
>
> Thank you for the review! Let us comment on the requested changes.
> 1. Our Bayesian estimator for semantic entropy works better than other estimators for two reasons. First, it leverages a small training set to learn the prior. Other estimators don't do this because, as they are not Bayesian, they do not have the concept of the prior. Note that the size of the training set does not scale with the number of queries to the estimator at inference time, hence the cost becomes entirely amortised. Second, the Bayesian estimator combines all evidence (the probabilities of generated sequences and the meanings) in an optimal way, based on the Bayes rule. Other estimators don't do this.
> 2. Any two consistent estimators will converge to the same value of semantic entropy (and hence AUROC) for large N. However, there are two caveats. First, this might require N much greater then 10. Second, not all estimators we compare to are consistent (for example, the rescaled estimator with heuristic rescaling of sequence probabilities isn't). We compare to these approaches because other researchers have used them, not because they necessarily have well understood theoretical properties.
>
> We will provide an updated version of the paper, making the above points more clear.

---

### Review · Reviewer_JXMa · 2025-06-03

**Summary Of Contributions:**

The paper addresses **hallucination detection**, a setup that requires identification of a model when it generates factually incorrect responses. One existing representative way to do this is to estimate the semantic entropy (aka. uncertainty) to detect hallucinations. On top of existing semantic entropy estimation, this paper proposes to use Bayesian approach, which performs better on hallucination detection in a more effective and efficient way. To validate their approach, they compare their design against baseline on multiple language models (LLaMA, Mistral) and showcase good improvements over existing approaches.

**Audience:**

Yes

**Broader Impact Concerns:**

The paper does not include a Broader Impact Statement section.

**Claims And Evidence:**

Yes

**Requested Changes:**

Please see weakness 1.

**Strengths And Weaknesses:**

### Strengths

**Comparisons**. Authors have made sufficient efforts in comparisons against other types of entropies and justify the efficacy of bayesian approach.
**Overall Results**. The experimental results well justify their claims and validate the efficacy of Bayesian approach for hallucination detection.

### Weaknesses

**Chosen models**. I notice that authors selects smaller size models to validate their approach (e.g., Llama-3.2-3B-Instruct, Mistral-Small-24B-Instruct-2501, whose activating parameters are small). It is frequent that conclusion made on small models cannot be generalized to larger models. Can authors add more models to further validate the bayesian approach?

---

> ### Author Response · Authors · 2025-06-08
> **Author Response.**
>
> Thank you for the review! We are a small lab and don't have the resources needed to run many experiments with very large models.
>
> However, we are able to run one new experiment on a model with 70 billion parameters (Llama-3.3-70B-Instruct). We will include it in the revised version of the paper.

---

### Author Response · Authors · 2025-06-08
**Response to All Reviews.**

We wanted to take the opportunity to thank all reviewers for their efforts in judging our work. Based on the feedback, we will provide an updated version of the paper by the end of next week at the latest.

---

### Author Response · Authors · 2025-06-11
**Revised Version of the Paper**

We wanted to once again thank all three reviewers for our work. We have now uploaded a revised version of the paper. We describe the changes below.

Reviewer JXMa:
- We added a new experiment with a 70 billion model (TriviaQA with Llama-3.3-70B-Instruct). Our method works very well on this large LLM.

Reviewer jKjg:
- We added a paragraph explaining why the Bayesian model works better (at the end of section 3.4).
- We added a note about consistency (what happens for large N) at the end of section 6.2.

Reviewer iPCg:
1. We added an explanation of the relationship between semantic entropy and hallucinations (box on page 2).
2.  We provide a self-contained description of the experimental setup (appendix B).
3. We correctly labelled the tables (they measure AUROC).
4. We did not provide an analysis of estimator performance as a function of response length because it would be orthogonal to our main line of work (which is doing the same as the paper by Farquhar et al. (2024) but with better sample efficiency). If you really think this is crucial, please do get back to us and we will find a way to include an experiment of this kind.

In addition:
- We added a note about consistency (what happens for large N) at the end of section 6.2.
- We added a sentence about novelty to the description of the Bayesian estimator. We stress that the idea of conditioning the entropy estimate on censored class probabilities, is, as far as we can tell, novel. It is also crucial to obtaining good empirical performance.

In addition, wanted to ask the Action Editor and the reviewers to consider endorsing our paper for presentation at ICLR. We believe hallucination detection is an important topic of interest there and the ICLR community would benefit from the message of our paper.

---

### Decision · Action_Editor_oKST · 2025-07-16

**Recommendation:** Accept as is

**Additional Comments:**

Reviewers were mainly concerned by the empirical results (in particular the role of the number of samples), the models used by the authors to validate their methodology and required clarifications with respect to previous works. The authors added an example with a large LLM (TriviaQA with Llama-3.3-70B-Instruct) which confirms their initial claims using larger models.
They also clarified the interest of their Bayesian approach in the context of Hallucination detection. While it is not surprising that Bayesian estimation outperforms maximum likelihood-based alternatives,  the application to hallucination detection may have interesting practical impact.
In addition, the proposed empirical evaluation of the influence of N (which is a key hyperparameter) is important to motivate the use of this Bayesian approach. Overall, these improvements of the paper, combined with the numerical results which suggest that the Bayesian estimator mostly outperforms the alternatives and provides interesting results for small N provide a convincing practical contribution.

The authors could carefully proofread the paper and extend the conclusion, for instance by providing limitations and perspectives on more advanced Bayesian approach to improve sample efficiency.

**Audience:**

Yes

**Audience Explanation:**

Detecting hallucinations in LLMs is a task of great practical interest and the authors based their work on state-of-the art practical results. The contribution lies therefore in an area of potential interest for a large audience in the machine learning community.

**Claims And Evidence:**

Yes

**Claims Explanation:**

This paper focuses on detecting hallucination of large language models using semantic entropy. They introduce a Bayesian framework to extend the work of  (Farquhar et al., 2024), to obtain an estimate of the semantic entropy. Their aim is to improve in particular the performance of previous estimators based on Monte Carlo sampling which are too computationally intensive and histogram-based approaches. The proposed algorithm can be used without a prior knowledge on the number of meanings associated with the generated sentences by the LLM given the proposed context.

The numerical experiments highlight that the method requires less samples from the model to achieve better hallucination detection metrics than the baselines. The authors assess the performance of their estimator using a variety of language models which supports the empirical claims of the submission.
In addition, the authors proposed during the rebuttal empirical evaluation of the influence of the number of samples to assess consistency, which is a key hyperparameter of the approach (using either a fixed budget of samples per prompt or an adaptive number of samples per prompt).